# A unified connectomic target for deep brain stimulation in obsessive-compulsive disorder

Ningfei Li [1]✉, Juan Carlos Baldermann[2], Astrid Kibleur[3,4], Svenja Treu[5], Harith Akram [6,7], Gavin J. B. Elias[8], Alexandre Boutet [8,9], Andres M. Lozano[8], Bassam Al-Fatly [1], Bryan Strange [5], Juan A. Barcia[10], Ludvic Zrinzo [6,7], Eileen Joyce [6,7], Stephan Chabardes[3], Veerle Visser-Vandewalle[11], Mircea Polosan[3,12,13], Jens Kuhn[2,14], Andrea A. Kühn[1] & Andreas Horn [1]

Multiple surgical targets for treating obsessive-compulsive disorder with deep brain stimulation (DBS) have been proposed. However, different targets may modulate the same neural network responsible for clinical improvement. We analyzed data from four cohorts of patients ($N = 50$) that underwent DBS to the anterior limb of the internal capsule (ALIC), the nucleus accumbens or the subthalamic nucleus (STN). The same fiber bundle was associated with optimal clinical response in cohorts targeting either structure. This bundle connected frontal regions to the STN. When informing the tract target based on the first cohort, clinical improvements in the second could be significantly predicted, and vice versa. To further confirm results, clinical improvements in eight patients from a third center and six patients from a fourth center were significantly predicted based on their stimulation overlap with this tract. Our results show that connectivity-derived models may inform clinical improvements across DBS targets, surgeons and centers. The identified tract target is openly available in atlas form.

[1] Charité – Universitätsmedizin Berlin, corporate member of Freie Universität Berlin, Humboldt-Universität zu Berlin, and Berlin Institute of Health, Movement Disorders and Neuromodulation Unit, Department for Neurology, Charitéplatz 1, 10117 Berlin, Germany. [2] Department of Psychiatry and Psychotherapy, Department of Neurology, University of Cologne, Medical Faculty, Cologne, Germany. [3] Univ. Grenoble Alpes, 38000 Grenoble, France. [4] OpenMind Innovation, 75008 Paris, France. [5] Laboratory for Clinical Neuroscience, Centre for Biomedical Technology, Universidad Politecnica de Madrid, Madrid, Spain. [6] Department of Clinical and Movement Neurosciences, UCL Queen Square Institute of Neurology, London, UK. [7] National Hospital for Neurology and Neurosurgery, UCL Queen Square Institute of Neurology, London, UK. [8] University Health Network, Toronto, ON, Canada. [9] Joint Department of Medical Imaging, University of Toronto, Toronto, ON, Canada. [10] Hospital Clínico San Carlos, Neurosurgery Department, Universidad Complutense de Madrid, Madrid, Spain. [11] Department of Stereotactic and Functional Neurosurgery, University of Cologne, Cologne, Germany. [12] Inserm, U1216, Grenoble Institut des Neurosciences, 38000 Grenoble, France. [13] Psychiatry Department, CHU Grenoble Alpes, 38000 Grenoble, France. [14] Department of Psychiatry, Psychotherapy and Psychosomatics, EVKLN, Johanniter Hospital Oberhausen, Oberhausen, Germany. ✉email: ningfei.li@charite.de

Obsessive-compulsive disorder (OCD) is a debilitating disease with a life-time prevalence of around 2.3%[1]. Treatment of severe cases by deep brain stimulation (DBS) to the ALIC has been approved by the FDA (Humanitarian Device Exemption) in 2009[2]. A variety of other targets have been proposed, however, including the STN[3,4], nucleus accumbens (NAcc)[5], ventral capsule/ventral striatum (VC/VS)[6], inferior thalamic peduncle (ITP)[7], bed nucleus of the stria terminalis (BNST)[8], anteromedial globus pallidus interna (amGPi)[9], superolateral branch of the medial forebrain bundle (slMFB)[10] and medial dorsal and ventral anterior nuclei of the thalamus (MD/vANT)[11] (for an overview see ref. [12]). A recent prospective clinical trial implanted four electrodes per patient with one pair in the STN and one in the ALIC[13].

In parallel, DBS has experienced a conceptual paradigm-shift away from focal stimulation of specific brain nuclei (such as the subthalamic nucleus or globus pallidus in Parkinson's disease; PD) toward modulating distributed brain networks (such as the motor basal-ganglia cortical cerebellar loop in PD)[10,14–17]. Although the concept of modulating white-matter tracts (instead of gray matter nuclei) is certainly not new (and anterior capsulotomy was introduced in the ~1950s by Talairach and Leksell[18]), novel MRI technologies such as diffusion-weighted imaging-based tractography are now increasingly used in functional neurosurgery in order to more deliberately target white-matter tracts[16]. In this translational development, the Coenen and Mayberg groups should be explicitly mentioned, among others, for pioneering and rapidly translating the use of tractography to functional surgery since around 2009[10,14,15,19].

It could be possible that, of the multiple targets proposed, some —or most—may in fact modulate the same brain network to alleviate symptoms. Such a concept has been proposed in the past by Schlaepfer and colleagues for the case of treatment-refractory depression[20]. Namely, the superolateral branch of the medial forebrain bundle may connect most if not all surgical targets that were proposed for treatment of depression (e.g. subgenual cortex, ALIC, NAcc, habenula). Thus, in theory, the tract itself could be a surgical target—and could be modulated in a similar way when targeting various points along its anatomical course. Accordingly, already, Coenen et al.[10] surgically implanted electrodes to directly target this tract instead of a localized target, also in OCD. The tract connected the ventral tegmental area and the prefrontal cortex and authors referred to it as the superolateral branch of the medial forebrain bundle.

Other invasive therapies, such as cingulotomy and capsulotomy also aimed at disrupting connectivity from frontal regions by lesioning white-matter bundles[21]. It could recently be shown that such tract- or network-based concepts may be used to predict clinical improvements across DBS centers and surgeons for the case of Parkinson's disease[22,23]. Based on modern neuroimaging methods and high-resolution connectomic datasets, connectivity of DBS electrodes to specific cortical regions was associated with stronger therapeutic effects in various diseases treated with this surgical procedure[22,24–26].

For the case of OCD, Baldermann et al.[24] recently demonstrated that structural connectivity from DBS electrodes to medial and lateral prefrontal cortices was associated with stronger symptom alleviation. Crucially, they were also able to identify a specific subsection of the ALIC that was highly associated with symptom improvements after one year of DBS. Of note, connectivity to this fiber tract was able to predict ~40% of the variance in clinical outcome in out-of-sample data. The bundle connected to both medial dorsal nucleus of the thalamus and to the anterior part of the STN (which have received substantial attention in the context of OCD). The STN itself is a prominent target for DBS of various diseases including PD, dystonia, OCD and Tourette's syndrome[27]. The small nucleus receives widespread direct afferents from most parts of the prefrontal cortex and is involved in motor, associative and limbic processing[28]. Due to these spatially organized cortico-subthalamic projections, the nucleus has functional zones that largely follow the organization of the frontal cortex, i.e. sensorimotor parts of the STN are situated posterior, followed by pre-/oculomotor-, associative and limbic domains in anteromedial direction.

Consequently, the anterior (associative/limbic) parts of the STN have been targeted by DBS for OCD[29]; these same anterior subregions were exclusively connected to the tract target identified by Baldermann et al.[24] in ALIC-DBS patients. Following up on this, our present study aimed at testing whether the same tract could be associated with good clinical outcome in a cohort treated with STN-DBS. We retrospectively analyzed two cohorts of DBS patients that were treated with either STN-DBS or ALIC-DBS in order to test our hypothesis, that the same tract could potentially predict clinical improvement in STN-DBS as well as ALIC-DBS. In this attempt, we identified a common tract that already became apparent when analyzing either cohort alone. After calculating the tract exclusively based on data of one cohort (e.g. ALIC), we cross-predicted outcome in the other cohort (e.g. STN), and vice versa. We then tested predictive utility of this tract in two additional cohorts from a third and fourth center. Finally, we set the resulting tract target into the larger context of OCD-DBS literature and tested, whether it could be used to explain outcomes of reported clinical studies with different surgical targets.

## Results

**Clinical results.** Two cohorts (Cologne; ALIC target; $N = 22$; and Grenoble; STN target; $N = 14$, two electrodes in each patient) formed a training and cross-validation sample in which the tract target was identified and validated. Each of the two cohorts were first analyzed independently, then used to cross-predict outcome in patients from the other one. The main part of our analyses focuses on these two cohorts. As further validation of results, two additional test cohorts were included (Madrid: two electrodes in each patient targeting bilateral nucleus accumbens (NAcc); London: four electrodes in each patient targeting bilateral ALIC and STN).

Patients in all cohorts were of similar age with a similar Y-BOCS score at baseline and comparable Y-BOCS improvement scores (Table 1). In the first test cohort (Madrid; NAcc target; $N = 8$), improvement scores were taken after activating each of the four electrode contact pairs for 3 months, respectively (following the clinical protocol described in ref. [30]). This resulted in a total of 32 data points. In the second test cohort (London; both ALIC and STN target; $N = 6$, four electrodes in each patient), stimulation parameters resulted from an optimized phase following parameter optimization.

Electrode localization confirmed accurate placement to each of the three target regions for all patients of the four cohorts (Fig. 1).

**Connectivity analysis.** Connectivity analysis results seeding from electrodes of the two training cohorts (Cologne and Grenoble) based on the $N = 985$ HCP normative connectome are shown in Fig. 2. The overall connectivity of electrodes to other areas in the brain (without weighing for clinical improvement) was strikingly different between the two cohorts (Fig. 2, top row). This is hardly surprising as it mainly reflects the overall structural connectivity profiles of the two DBS targets. The STN as a widely connected basal-ganglia entry point and the ALIC as a white-matter structure are differently connected in the brain. However, when tracts were weighted by their ability to discriminate between good and poor responders (using the fiber $T$-score method described

**Table 1 Patient demographic details and clinical results of the two cohorts.**

| | ALIC-DBS cohort (mean ± SD) | STN-DBS cohort (mean ± SD) | NAcc DBS cohort (mean ± SD) | Combined DBS cohort (mean ± SD) |
|---|---|---|---|---|
| Center | University Hospital Cologne | University Hospital Grenoble | Hospital Clínico San Carlos Madrid | University Hospital London |
| Reference(s) | [22, 31] | [28] | [40] | [16] |
| N of patients (females) | 22 (12) | 14 (9) | 8 (4) | 6 (1) |
| N of electrodes | 44 | 28 | 16 | 24 |
| Age | 41.7 ± 20.5 | 41 ± 9 | 35.3 ± 10.4 | 45.5 ± 10.5 |
| Y-BOCS baseline | 31.3 ± 4.4 | 33.4 ± 3.7 | 30 ± 7.75 | 36.2 ± 1.8 |
| Y-BOCS after DBS | 20.7 ± 7.7 (12 months postop) | 19.6 ± 10.6 (12 months postop) | 14.75 ± 7.2 (3 months postop of best contact) | 14.3 ± 4.1 (optimized phase in ref. [16]) |
| Absolute Y-BOCS Improvement | 9.6 ± 6.5 | 13.8 ± 10.8 | 15.1 ± 9.6 | 21.83 ± 5.7 |
| % Y-BOCS Improvement | 31.0 ± 20.5% | 41.2 ± 31.7% | 47.8 ± 23 | 50.0 ± 12.6% |

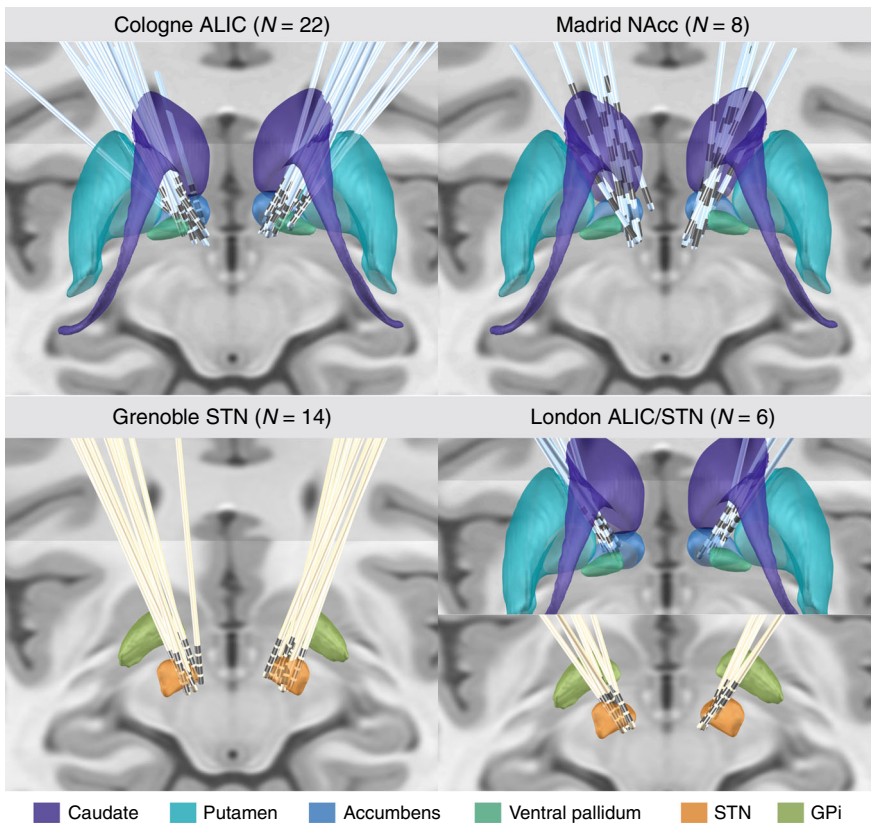

**Fig. 1 Overview of lead electrode placement.** The two training/cross-validation cohorts (left) targeting ALIC (Cologne) and STN (Grenoble), and the two test cohorts (right) targeting NAcc (Madrid) and both ALIC & STN with four electrodes per patient (London) are shown. Subcortical structures defined by CIT-168 Reinforcement Learning Atlas[63] (ALIC/NAcc region) and DISTAL Atlas[64] (STN region), with coronal and axial planes of the T1-weighted ICMB 152 2009b nonlinear template as background.

below), a positively discriminative tract to the medial prefrontal cortex emerged in each cohort even when cohorts were analyzed independently (Fig. 2, middle row). The degree of lead connectivity to this tract correlated with clinical improvement ($R = 0.63$ at $p < 0.001$ in the ALIC cohort and $R = 0.77$ at $p < 0.001$ in the STN cohort; Fig. 2, bottom row).

Of note, these correlations are somewhat circular and meant to describe the degree of how well discriminative tracts could explain the same sample of patients on which they were calculated. More interestingly, in the next step, the tract was calculated exclusively on data from the STN cohort and then used

to explain outcome in the ALIC cohort ($R = 0.50$ at $p = 0.009$) and vice versa ($R = 0.49$ at $p = 0.041$; Fig. 3).

Crucially, some VTAs of the ALIC cohort resided entirely below the identified tract and thus received a fiber $T$-score of (near) zero (also see blue example patient in Fig. 3, bottom right). The same holds true when either calculating the tract based on the STN cohort (Fig. 3) or the ALIC cohort itself (Fig. 2). To further investigate this matter, two-sample $t$-tests between improvements of patients with near-zero scores (fiber $T$-scores below 50) and the remaining patients with VTAs covering the tract well (scores above 50) were calculated. This showed that

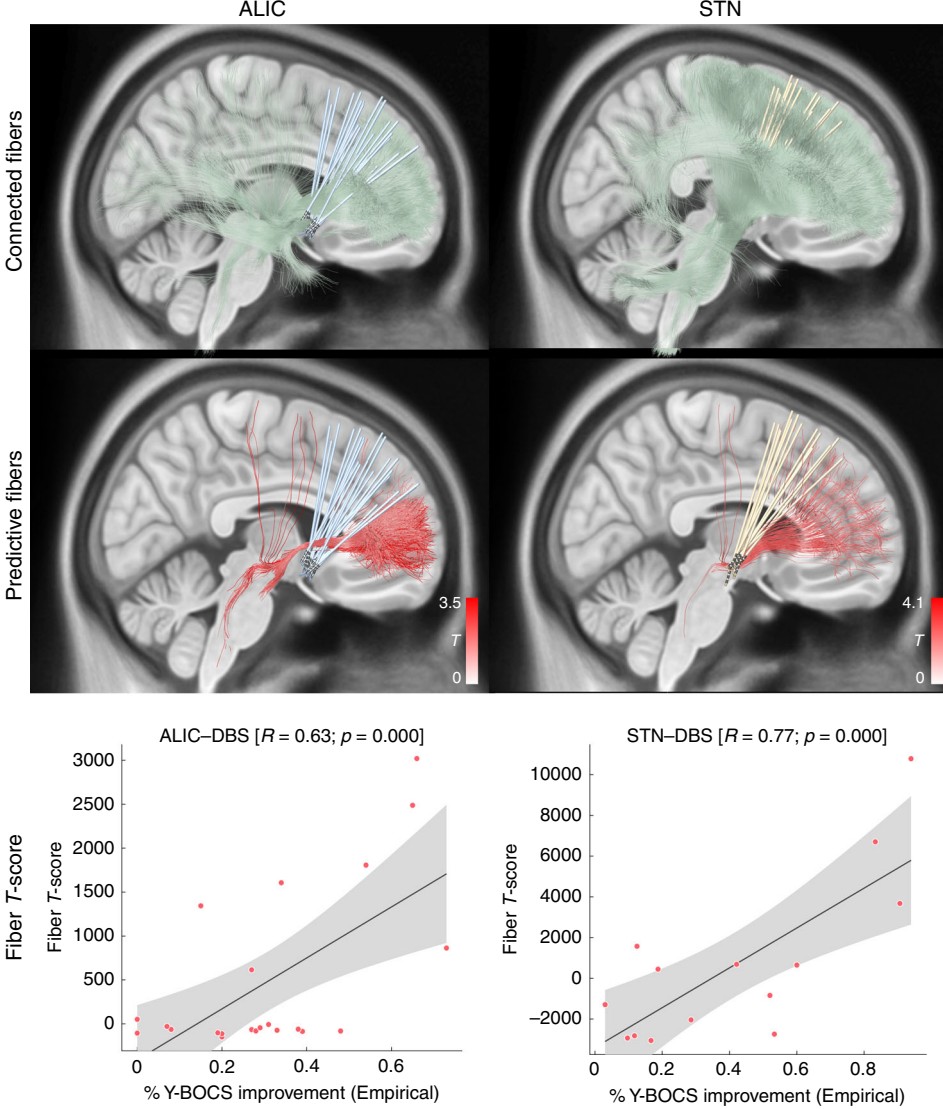

**Fig. 2 Predictive fiber tracts in training cohorts.** Top: all fibers connected to the sum of volumes of tissue activated (VTAs) of each cohort are shown in green. Middle: predictive fibers positively associated with clinical improvement are shown in red. Only positive fibers are shown here for reasons of clarity. See Fig. 3 for negatively associated tracts. The top 20% predictive fibers are displayed. Bottom: correlations between the degree of stimulating positively predictive tracts (sum of aggregated fiber $T$-scores under each VTA) and clinical improvements. Gray shaded areas represent 95% confidence intervals. This analysis is based on a normative connectome, a replication of it based on anatomically predefined pathways is shown in Supplementary Fig. 1.

VTAs with large overlap with the tract resulted in significantly better clinical improvement ($T = 6.0$ at $p < 10^{-5}$ when the tract was calculated on the ALIC cohort, Fig. 2, and $T = 3.7$ at $p < 0.005$ when it was calculated on the STN cohort, Fig. 3).

Depending on the target, the analysis revealed different proportions of "positive" and "negative" fibers (ALIC cohort: 22.2k positive vs. 1.9k negative fiber tracts selected from the group connectome; STN cohort: 45.1k positive vs. 48.6k negative fibers and both cohorts combined: 54.4k positive vs. 9.6k negative fibers).

**Replication on independent test cohorts.** In the next step, the analysis was performed on the two cohorts combined. Again, the same tract emerged, now even more clearly (Fig. 4, top). Bundles were selected from the connectome and visualized, that were predominantly connected with VTAs of patients from both cohorts with good (red) or poor (blue) improvement, respectively. The resulting positive discriminative tract traversed slightly dorsal to the group of electrodes of the ALIC cohort and coursed centrally or slightly ventral to the electrodes of the STN cohort. This

tract was then used to predict outcome in two independent test cohorts of patients that underwent surgery in a third and fourth center (Madrid & London; Fig. 4, bottom). Although the surgical target of the Madrid cohort was the NAcc, electrode placement was comparable to the ALIC/Cologne cohort (Fig. 1). Here, improvements were taken for each contact pair that had been switched on during a 3-month interval, leading to 32 data points (Fig. 4, bottom left, active contact pair color coded). In the London cohort, patients had received two electrodes to each target (four in total) and fiber $T$-scores scores were summed up across targets. In both test cohorts, stimulation overlap with the tract target significantly correlated with empirical improvement (Madrid: $R = 0.50$ at $p < 0.001$, London: $R = 0.75$ at $p = 0.040$). Of note, VTAs in the London sample were estimated with a different software (see Methods), patients received four electrodes and the clinical scores represented an "optimized" phase following 9 months of a clinical trial[13].

Given the high amount of false-positive connections present in dMRI-based connectomes[31], we replicated all findings of the

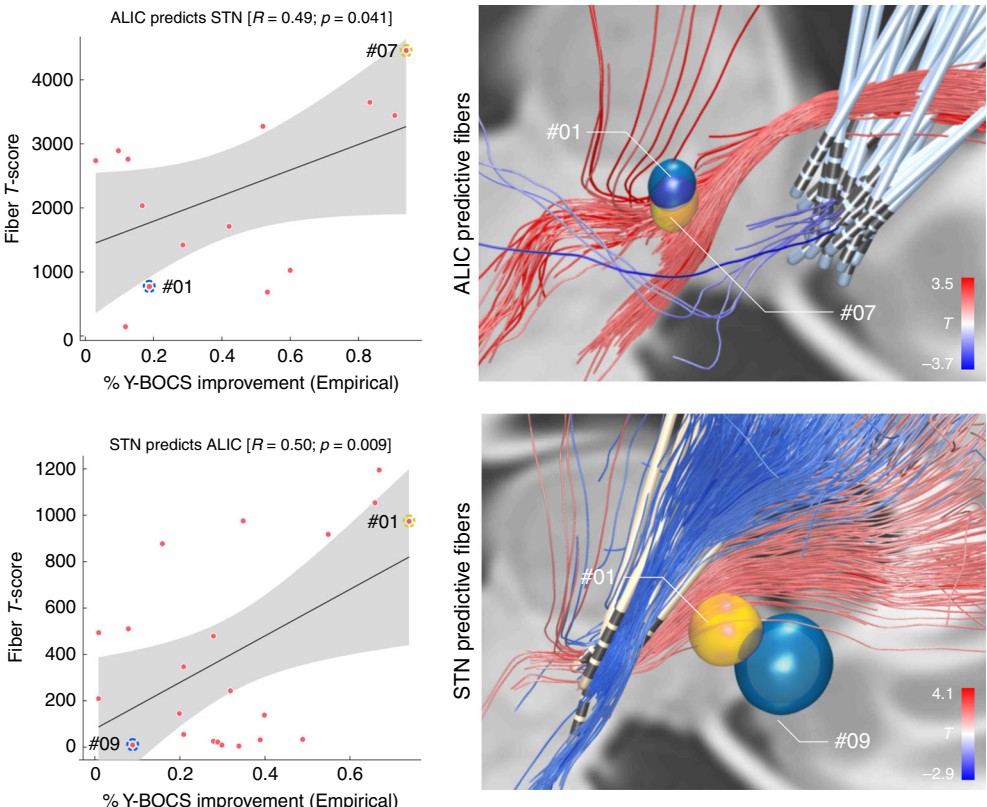

**Fig. 3 Cross-prediction between ALIC and STN training cohorts.** Top: when the tract was calculated exclusively based on data from the ALIC cohort, it was used to calculate fiber *T*-scores for all patients in the STN cohort. These were correlated with clinical improvements in the STN cohort. One example patient with strong overlap of the tract (yellow) received a high fiber *T*-score, whereas one with less overlap received a lower score (blue). The two example patients are marked in the correlation plot on the left. Bottom: here, the tract was calculated exclusively on data from the STN cohort to predict outcome in patients in the ALIC cohort. Again, two example patients are shown. Gray shaded areas in the correlation plots represent 95% confidence intervals. Of note, here, some VTAs barely overlapped with the tract (as the blue example VTA) and consequently received a near-zero score.

study using a synthesized anatomical atlas that is based on established anatomical knowledge[17] and thus free of such false-positive connections. Results were highly similar and identified the hyperdirect pathway connecting the dorsal anterior cingulate cortex (dACC) to the STN to be most associated with clinical outcome (Supplementary Figs. 1 and 2).

**A tract target for OCD-DBS.** The tract target identified here may potentially "unify" some aspects of the STN and ALIC/NAcc targets for OCD. Thus, in a final analysis, we aimed at setting it into context with other DBS targets that were used in OCD-DBS, before. To do so, we converted literature-based targets into template space[32] and set them into relation with the tract target (see Fig. 5, Table 2 and Supplementary Methods). A large number of reported DBS targets for OCD seemed to cluster on or around the tract. Furthermore, clinical improvement values that had been reported in these studies could be significantly accounted for by calculating the weighted overlap between stereotactic target sites and the tract (Fig. 5c, see Supplementary Methods for details).

Given the potential clinical importance of the identified tract, we estimated a final version of the tract based on all four cohorts and characterized its anatomical properties using additional views relative to anatomical landmarks (Fig. 6 and Supplementary Fig. 3). Anatomically, the tract is a subpart of the well-characterized ALIC that connects areas of the prefrontal cortex with the subthalamic nucleus and MD nucleus of the thalamus[33,34]. Anatomical validity of the isolated tract was discussed with four anatomists and further experts in the field (see Acknowledgements section). In the motor domain, the

"hyperdirect pathway", i.e., a direct connection from frontal cortex to subthalamic nucleus, has been well established[35,36], functionally, but the STN is known to receive widespread and direct input from widespread areas of the prefrontal cortex[33]. Thus, the main part of the specific bundle delineated here may represent a route of direct input from frontal regions to the STN. In addition, connections between mediodorsal nucleus of the thalamus and prefrontal regions received slightly lower (but positive) *T*-scores and are not shown in 3D visualizations but well visible in 2D sections shown in Fig. 6. The bundle most negatively associated with clinical improvement was the posterior limb of the anterior commissure, connecting bilateral temporal cortices.

To properly define the anatomical course of this tract, we openly released it as an atlas in stereotactic (MNI) space within Lead-DBS software (www.lead-dbs.org). Of note, Lead-DBS is scientific and not clinical software and the tract should not be vacuously used for any form of clinical decision making[37].

## Discussion

We analyzed data from four cohorts of OCD patients with different DBS targets using a connectomic approach. Strikingly, the same optimal tract target emerged when separately analyzing either an ALIC-DBS or STN-DBS cohort, alone. Among other regions, this bundle connected dorsal anterior cingulate and ventrolateral prefrontal cortices to the anteriomedial STN. When the tract was calculated on either cohort alone, it could be used to cross-predict clinical improvement in the other cohort, respectively. Furthermore, variance in clinical outcomes in two independent test cohorts from a third and fourth center could be

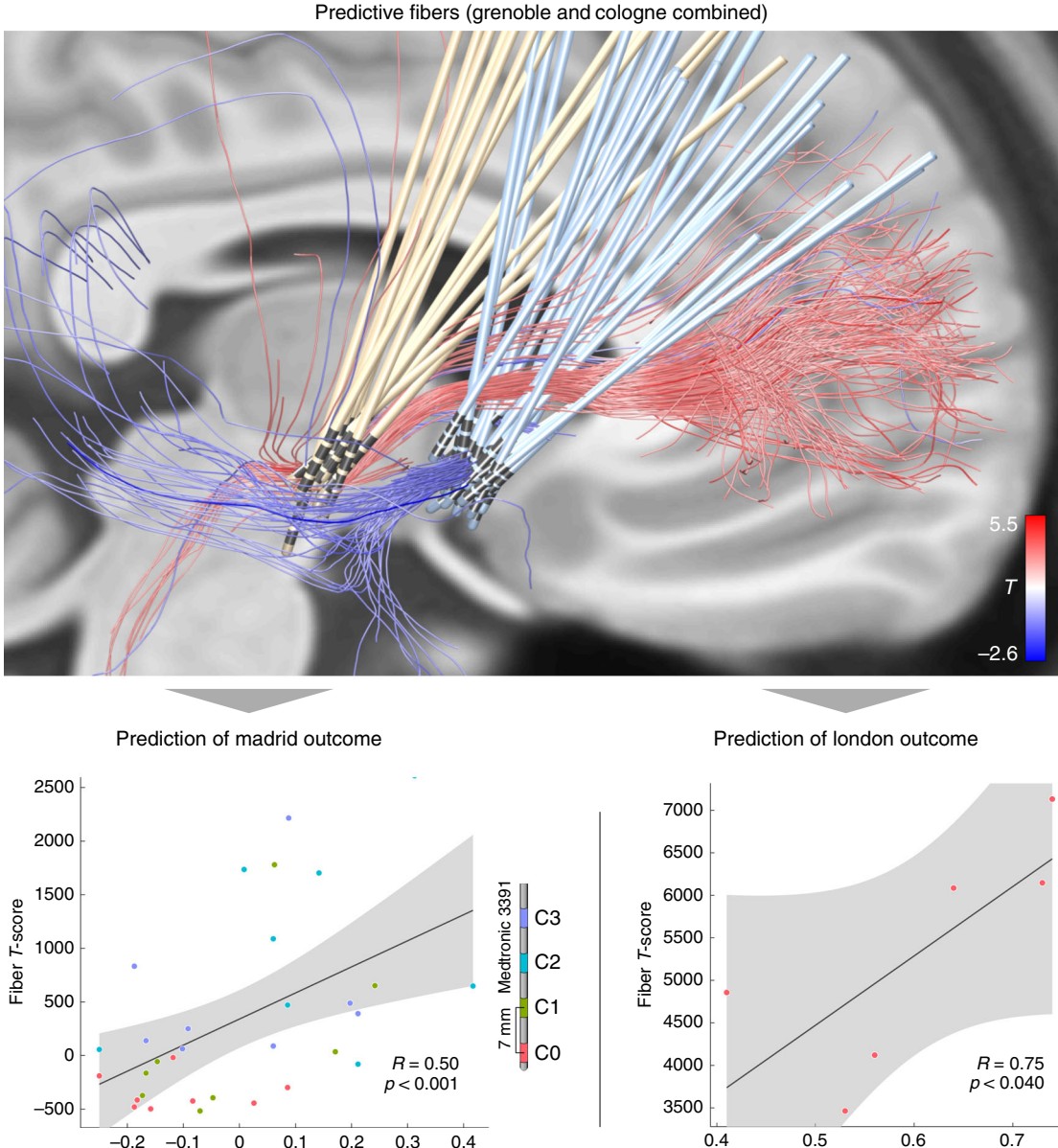

**Fig. 4 Predictions in test cohorts.** Top: predictive fibers calculated on both training cohorts (Cologne & Grenoble) irrespective of their target. Red fibers are positively associated with clinical improvement, blue fibers negatively. Bottom: the sum of aggregated fiber $T$-scores under each VTA predicted %-Y-BOCS improvements in eight patients with four settings each ($N = 32$ stimulations) of the Madrid cohort (left) and six patients of the London cohort with dual stimulation (four electrodes) of STN and ALIC (right). Gray shaded areas represent 95% confidence intervals. Please note that p-values in this manuscript are based on random permutation testing. Based on classical tests, the result shown in the lower right panel would remain significant in a one-sided test, only ($p$-one-sided $= 0.044$, $p$-two-sided $= 0.089$). A replication of this result based on anatomically predefined pathways is shown in Supplementary Fig. 2.

significantly predicted based on stimulation overlaps with the tract. Finally, literature-based stimulation sites for OCD seemed to cluster close to the identified tract. Indeed, their spatial proximity to the tract correlated with reported clinical improvements across studies.

The subthalamic nucleus receives afferents from a large portion of the prefrontal cortex by hyperdirect pathways that are known to traverse within the internal capsule[33,38]. In rodents, lesions to such a "limbic hyperdirect pathway" led to diminished discriminative accuracy and increased perseveration[39]. One classical cortical region, which was described as an origin of limbic hyperdirect input is the dACC[17,33,40], which has a prominent role in the classical cortico-striato-thalamo-cortical (CSTC)

model of OCD[40] and leads to improvement of OCD symptoms when directly lesioned in humans[41]. The normative connectome analysis identified the dACC as a cortical connection site to the identified tract, among others. Because of the high amount of false-positive connections in diffusion MRI-based connectomes[31,42], we repeated the analysis using an atlas of predefined anatomical tracts[17]. Here, the hyperdirect pathway connecting dACC to the STN was isolated as the only of five bundles in the ALIC that were included in the atlas (Supplementary Figs. 1 and 2). Thus, hyperdirect cortical input from dACC to STN could be an anatomical and functional substrate of the identified bundle. In this context, it is crucial to note that the atlas by nature cannot represent each and every white-matter

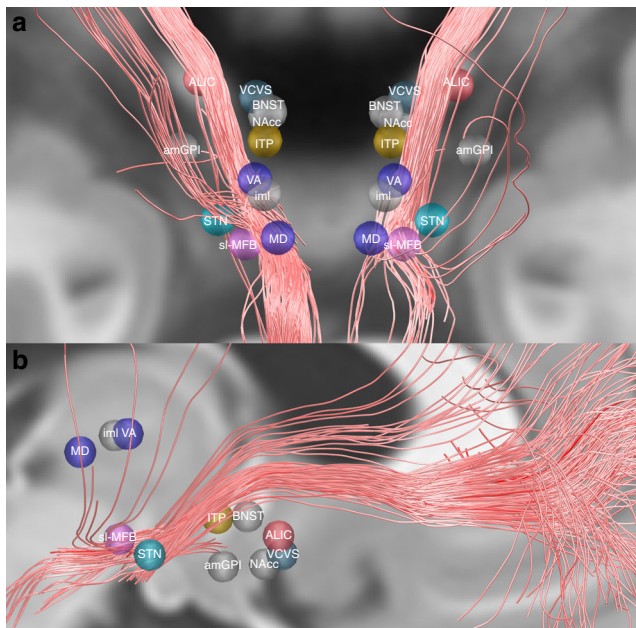

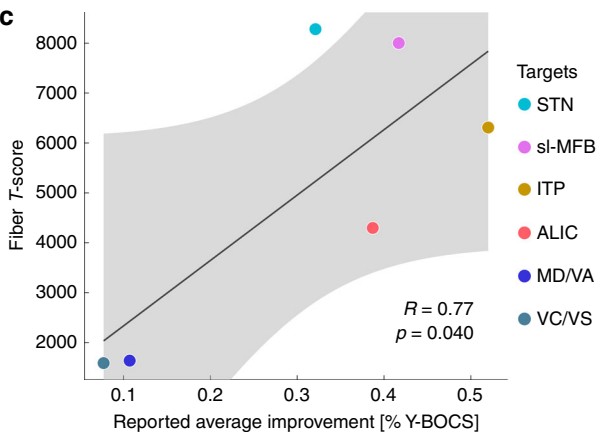

**Fig. 5 Literature defined OCD targets in relationship to the identified tract.** Overview of the positively predictive fiber tracts identified in the present study are shown in synopsis with DBS targets for treatment of OCD from reported studies. Note that most targets were reported for the tip of the electrode, thus, active stimulation may have occurred dorsal to shown targets (Table 2). **a, b** Reported average targets mapped to standard space. **c** The degree of weighted overlap between stimulation sites and the identified tract. These were correlated with reported average %-Y-BOCS improvements of published studies (where available, other sites marked in gray; see Supplementary Methods for details). Gray shaded area represents 95% confidence intervals.

bundle that exists in the ALIC / STN region and shows "gaps" in between the included bundles (Supplementary Figs. 1 and 2). Thus, while normative connectomes include a large number of false-positive fibers, the atlas may instead be prone to false-negative connections, as some tracts are simply not included. For instance, it is known that the STN receives direct input from other areas of the prefrontal cortex such as the ventrolateral prefrontal cortex[43]. In summary, although dACC and vlPFC are likely candidates to play a functional role, our methods and results are unable to determine the exact cortical region(s) of origin with absolute certainty. Despite this limitation, our results define a precise three-dimensional reconstruction of the tract itself (i.e. a definition of where it exactly traverses within the ALIC) in standard stereotactic space.

A highly similar pathway that already served as a tract target in an $N = 2$ case-series of OCD patients[10] also traversed within the ALIC but has instead been referred to as the superolateral branch of the medial forebrain bundle (slMFB)[37]. The original anatomical definition of the medial forebrain bundle suggests a more ventral route connecting the ventral tegmental area to the olfactory cortex while bypassing the red nucleus laterally[34]. In other words, the anatomical definition of the medial forebrain bundle does not traverse within the ALIC. This mismatch between the surgical target (slMFB) and anatomical literature (mfb) has recently been confirmed by the original authors of the surgical target and they now referred to it by vtaPP (for ventral tegmental area projection pathway)[44]. This potentially misleading nomenclature of the surgical slMFB target has suggested that results in two previous OCD studies would be conflicting, while anatomically, their results agreed. Both studies favored a similarly defined tract within the ALIC, which was referred to as slMFB in one study[45] and as anterior thalamic radiation in the second[24]. To readers, this suggested conflicting results while they were in fact confirmatory (based on the location of both tracts within the ALIC). Thus, we welcome the recent steps taken to move away from calling the surgical target slMFB toward calling it vtaPP[44]. This said, our interpretation of the identified tract differs. Our findings reveal a tract connecting frontal areas with the STN (cf. Supplementary Fig. 3C and results from the basal-ganglia pathway atlas, Supplementary Figs. 1 and 2). Thus, we attribute the tract to afferents of the STN (limbic hyperdirect pathway) as opposed to efferents of the ventral tegmental area implied by the term vtaPP[44].

This interpretation is supported by combined analyses of dMRI and tracing methods in nonhuman primates as well as human subjects, which were used to segregate prefrontal fibers passing through the internal capsule[46]. Fibers that originated from ventrolateral prefrontal cortices (areas 45 and 47) were shown to terminate in the medial part of the STN and the MD nucleus of the thalamus—precisely corresponding to the tract described here. Alternatively—or additionally—the hyperdirect pathway projecting from dACC to the STN may be functionally involved in mediating treatment outcome. As mentioned, a strong additional hint for this latter hypothesis is that lesions to the dACC itself have beneficiary effects on OCD[41].

Based on our results, two testable hypotheses with implications above and beyond OCD could be proposed. First, different surgical targets may reduce the same symptoms equally well—potentially by modulating the same tract or network. Second, in addition, they may modulate not only one (shared) network but other networks that are not shared, resulting in different changes across other behavioral domains. This can be seen by widely different connectivity profiles of the targets (Fig. 2, top row) and differential effects of STN vs. ALIC stimulation on depressive/cognitive functions described by Tyagi et al.[13]. Thus, one may speculate that networks are symptom-specific (and not disease-specific). When modulated, these networks or tracts seem to not ameliorate a specific disease but rather specific symptoms present in the disease.

In OCD, accordingly, different symptom types (for example contamination vs. checking) were found to activate different prefrontal sites (ventromedial vs. dorsolateral, respectively)[47]. Similar observations were made in other diseases, before. For instance, Akram and colleagues demonstrated that connectivity to specific cortical regions was associated with improvement in different clinical features of Parkinson's disease (e.g. connectivity to M1 preferentially reduced tremor while to the SMA reduced rigidity and bradykinesia)[25]. Similarly, connectivity from electrodes to M1 was associated with tremor improvement in Essential Tremor[48].

**Table 2 DBS targets for treatment of OCD defined in the literature.**

| DBS target | References | Number of patients | % Y-BOCS change | AC/PC coordinates | Relative to | Target type | MNI coordinates (Fig. 5) |
|---|---|---|---|---|---|---|---|
| STN | Mallet et al.[66] | 8 | 32.1 | NA | AC | Tip of the electrode | ±11.30 −9.90 −7.81 |
| amGPi | Nair et al.[9] | 4[a] | NA | ±14.47 9.85 −3.28 | MCP | Tip of the electrode | ±15.66 −1.41 −8.22 |
| VC/VS | Tsai et al.[67] | 1 | 7.7 | ±7.5 16.3 −3.05 | MCP | Tip of the electrode | ±7.92 5.51 −9.01 |
| slMFB | Coenen et al.[10] | 2 | 41.7 (at 12 months) | ±7.6 −1.72 −3.0 | MCP | Active contacts | ±8.35 −13.64 −7.00 |
| NAcc | Sturm et al.[5] | 4 | NA | ±6.5 2.5 −4.5 | AC | Tip of the electrode | ±6.98 3.69 −10.55 |
| ALIC | Nuttin et al.[68] | 6 | 38.7 | ±13 3.5 0 | AC | Tip of the electrode | ±13.84 5.17 −5.04 |
| MD | Maarouf et al.[11] | 4 | 10.7 | ±4.7 −18.52 4.87 | AC | Active contacts | ±5.10 −18.17 2.59 |
| VA | Maarouf et al.[11] | 4 | 10.7 | ±6.84 −13.76 7.78 | AC | Active contacts | ±7.52 −12.68 5.60 |
| iml | Maarouf et al.[11] | 4 | 10.7 | ±5.78 −14.9 7.08 | AC | Active contacts | ±6.36 −13.99 4.85 |
| ITP | Lee et al.[69] | 5 | 52.0 | ±6.5 −3 −0.5 | AC | Tip of the electrode | ±6.92 −1.84 −5.13 |
| BNST | Nuttin et al.[70] | 4 | NA | ±6 0 0 | AC | Tip of the electrode | ±6.33 1.39 −4.87 |

*MD* medial dorsal thalamic nucleus, *VA* ventral anterior thalamic nucleus, *iml* internal medullary lamina, *MCP* mid-commissural point, *AC* anterior commissure.
[a]Tourette patients, with prominent symptoms of OCD.

**Fig. 6 Anatomical course of discriminative fibers shown in MNI space.** The tract is connected to the subthalamic nucleus and mediodorsal nucleus of the thalamus, traverses through the anterior limb of the internal capsule and has a wide array of frontal connections including dorsal anterior cingulate cortex and ventrolateral prefrontal cortex. The tract most negatively associated with clinical improvement was the anterior commissure.

Supporting the first hypothesis, our study was able to predict symptom-specific clinical improvement across DBS targets and centers based on connectivity data. Although the tract that our data seems to shape out is predictive for Y-BOCS improvement, different tracts could have emerged when repeating the analyses for depressive or cognitive flexibility symptoms (as analyzed by Tyagi et al.[13]).

Going further, shared symptom networks could be present in other diseases for which multiple surgical targets are investigated. Major depression and Tourette's syndrome are obvious examples and extensive work in this direction is currently ongoing[14,49,50]. Similar concepts could even be applied to more established targets such as STN vs. GPi DBS[51] or symptom-specific alleviations across diseases.

Potentially, DBS surgery in the (distant) future could involve detailed preoperative phenotyping to establish a broad patient-specific symptom score. Based on databases of clinical improvements along affected symptom axes, a mix of networks that should be modulated to alleviate each patient's specific symptom profile could be identified. Such concepts are still mostly speculation but could be investigated in future studies. This said, we must emphasize that the present study investigated data on a group level and utilized connectivity data from individuals without OCD. As mentioned by others in the very context, we could not agree more that surgical decision making for DBS should not be based on such aggregated normative data, alone[37]. Further studies are required to determine whether individual patient connectivity or generic connectome data (or both) could assist with optimizations in surgical targeting or DBS programming by determining crossing sites of symptom networks for specific patients.

Several limitations apply for the current work. First and foremost, the retrospective character of the study is not ideal to compare and study effects of clinical outcome which is why we kept clinical information to a minimum and instead referred to the underlying clinical studies.

Second, it has been shown that dMRI-based tractography reconstructs a very high proportion of false-positive fibers in recent open challenges[31,42]. We aimed at reducing the risk of false-positive tractography results in four ways. First, we used the tracking method that achieved the highest (92%) valid connection score among 96 methods submitted from 20 different research groups in a recent open competition[31]. Second, we used highest quality multi-shell diffusion data[52] acquired on a high N (985 subjects) at a state-of-the-art imaging center (HCP data acquired at Washington University in St. Louis, see Acknowledgements). Third, we compared the tract results with anatomy text-books and discussed its validity with four anatomists (see Acknowledgements). Fourth, we replicated findings based on an atlas that is based on predefined anatomical tracts (Supplementary Methods). The tract described in the present study matches results from this atlas (Supplementary Figs. 1 and 2). However, the potential that the tract represents a false-positive result may not be completely ruled out given the fundamental limitations of dMRI-based tractography[31,42].

Third, we used normative connectome data instead of patient-specific diffusion-weighted MRI data (which is not available for most of the patients included). This poses marked limitations as such data cannot be representative of patient-specific anatomical variations. Still, we argue that some aspects about general pathophysiological mechanisms may be investigated using normative data and robust cross-validations across cohorts shown here suggest this holds true. Use of normative connectomes has been introduced in other clinical domains where patient-specific MRI data is unavailable, such as stroke[53,54] or transcranial magnetic stimulation[55]. In DBS, the technique has been applied before and has led to models that could be used to predict improvements in out-of-sample data[22,24]. In addition to the practical advantage of being applicable to cases where patient-specific data is lacking, normative data also has the theoretical advantage of better data quality. In the present case, a connectome dataset was derived from a high N of 985 subjects scanned under research conditions by a specialized imaging center[52]. It may be logistically challenging to acquire data of such quality in a clinical routine setting (e.g. pre-operatively) in individual patients but could be feasible in specialized centers. Still, studies have pointed out that tractography-based DBS targets pointed to coordinates that were sometimes >2 mm apart from each other when repeating analyses on test–retest scans of the same subject[56]. Similarly, variance introduced by single subject scans was too high to be useful in a test–retest study that aimed at creating clinically useful and robust thalamic DBS targets[57]. However, patient-specific connectivity can never be reconstructed when using normative connectomes. Thus, normative connectomes will likely not embody the final solution to the connectomic surgery framework and will be challenged by advances in MRI technology and algorithm developments. Potentially, as a step in-between, using combined information from normative and patient-specific connectomes could embody a promising strategy that should be explored, in the future.

Fourth, inaccuracies in lead localization may result from the approach of warping electrodes into common space as done here. To minimize this issue, we used a modern neuroimaging pipeline that has been scientifically validated in numerous studies and involved advanced concepts such as brain shift correction[58], multispectral normalization, subcortical refinement[58] and phantom-validated electrode localizations[59]. The normalization strategy that was applied was found to automatically segment the STN as precisely as manual expert segmentations[60] and each step of the pipeline was carefully assessed and corrected if needed by a team with long-standing expertise in this area[58,61]. Besides, both post-operative CT (33 patients) and post-operative MRI (17 patients) were used for electrode localization in the current dataset. Although studies have reported similar agreement between the results based on the two modalities, this might still lead to slight inconsistencies across patients. A larger dataset acquired with a homogeneous protocol would be ideal to validate our results, in the future.

Finally, given the correlative nature of the study, our findings may not differentiate between local and global effects. For instance, the tracts may have spuriously originated in the ALIC group because a more dorsal stimulation resulted with better clinical outcome. The congruency between results derived from STN- and ALIC-cohorts resulting in the same fiber bundle still suggests that the identified tract could play a causal role. However, such a claim would need to be confirmed e.g. using optogenetics or electrophysiology.

Four main conclusions may be drawn from the present study. First, we show that the overall connectivity profiles of STN- and ALIC-DBS electrodes project to largely different areas in the brain. Second, data in each target alone singled out the same fiber bundle that was associated with long-term improvement of OCD symptoms when modulated either at level of the STN or the ALIC. Third, we demonstrated that it is possible to cross-predict clinical improvement of OCD patients across DBS target sites (ALIC/STN) and centers (Cologne/Grenoble). Finally, we confirm results by predicting outcome in two additional cohorts from different centers (Madrid/London) and set results into context of published reports.

## Methods

**Patient cohorts and imaging**. Fifty OCD patients from four centers were retrospectively enrolled in this study, among them 22 patients from University Hospital of Cologne implanted for ALIC-DBS, 14 patients from Grenoble University Hospital who underwent STN-DBS surgery, 8 patients who received bilateral electrodes targeting the NAcc from Hospital Clínico San Carlos in Madrid and 6 patients who received electrodes to both STN and ALIC from the National Hospital for Neurology and Neurosurgery in London. The patients from Cologne, Grenoble and Madrid received two electrodes each ($N = 44$ patients with $N = 88$ electrodes), the six patients in the London cohort received four electrodes each ($N = 6$ patients with $N = 24$ electrodes). All patients from Grenoble were bilaterally implanted with DBS electrodes 3389, as were all but three patients from Cologne, who received type 3387 electrodes (Medtronic, Minneapolis, Minnesota, US). Patients from London received models 3389 to the STN and 3387 to the ALIC. Patients from Madrid received models 3391. All patients qualified for DBS surgery based on their diagnoses of treatment-resistant severe OCD[13,24,29]. Severity of OCD was assessed both pre- and postoperatively using the Yale-Brown Obsessive-Compulsive Scale (Y-BOCS). Post-operative assessment took place 12 months after surgery in Cologne, Grenoble and London cohorts. In case of the London cohort, this followed a four-step clinical trial (2 × 3 months blinded stimulation at one target followed by 6 months of stimulation at both targets, the last 3 months using clinically optimized parameters. For details see ref. [13]). In the Madrid cohort, each of the four contact pairs was activated for 3 months, with a 1-month wash-out period between trials and a 3-month sham period. In our analysis, this led to 32 data points (i.e. stimulation-based outcomes). Patients' demographic details are provided in Table 1. All patients gave written informed consent. The protocols were approved by the Ethics Committee of each center, respectively. The current study was further approved by the local ethics committee of Charité—University Medicine Berlin in accordance with the Declaration of Helsinki.

For all patients in the four cohorts, high-resolution structural T1-weighted images were acquired on a 3.0-Tesla MRI scanner, before surgery. Post-operative computer tomography (CT) was obtained in thirty-three patients after surgery to verify correct electrode placement, while 11 patients from the Grenoble cohort and the six London patients received post-operative MRI instead. Post-operative MRI parameters were as follows. Grenoble cohort: T1-weighted 3D-FFE scans were acquired on a 1.5 T Philips MRI scanner with a 1.0 × 1.0 × 1.5 mm³ voxel size; TR: 20 ms, TE: 4.6 ms, flip angle: 30 deg. London cohort: T1-weighted 3D-MPRAGE scans were acquired on a 1.5 T Siemens Espree interventional MRI scanner with a 1.5 × 1.5 × 1.5 mm³ voxel size and three-dimensional distortion corrected using the

**Fig. 7 Summary of methods to define a _T_-value for each tract. a** For each fiber, VTAs were grouped into either connected (C; yellow) or unconnected (UC; blue) sets across patients. **b** Two-sample _t_-tests between clinical improvements in connected and unconnected VTAs were calculated in a mass-univariate fashion for each fiber tract separately. **c** The resulting _T_-value of this analysis leads to the "weight" that each fiber received, as well as the color in visualizations throughout the manuscript. Here, red means that the fiber tract is favorably connected to good responders, whereas blue indicates the opposite (and the saturation of tracts denotes how discriminative they are).

scanner's built-in module; TR: 1410 ms, TE: 1.95 ms, FOV: 282 mm, flip angle: 10 deg, acquisition time 4 min and 32 s, relative SNR: 1.0.

**DBS lead localization and VTA estimation**. DBS electrodes were localized using Lead-DBS software (http://www.lead-dbs.org)[58]. Post-operative CT and MRI scans were linearly coregistered to preoperative T1 images using Advanced Normalization Tools (ANTs, http://stnava.github.io/ANTs/)[62]. Subcortical refinement was applied (as a module in Lead-DBS) to correct for brain shift that may have occurred during surgery. Images were then normalized into ICBM 2009b Nonlinear Asymmetric ("MNI") template space using the SyN approach implemented in ANTs, with an additional subcortical refinement stage to attain a most precise subcortical alignment between patient and template space ("Effective: Low Variance" preset as implemented in Lead-DBS). This specific method was top performer for subcortical image registrations in a recent comparative study that involved >10,000 nonlinear warps and a variety of normalization techniques[60]. Both coregistrations and normalizations were visually reviewed and refined, if needed. DBS electrodes were then localized using Lead-DBS and warped into MNI space.

In the Grenoble, Cologne and Madrid groups, VTA were estimated using a finite element method (FEM)[58]. A volume conductor model was constructed based on a four-compartment mesh that included gray matter, white matter, electrode contacts and insulated parts. Gray matter was defined by the CIT-168[63] and DISTAL[64] atlases for the ALIC-/NAcc and STN-cohorts, respectively. These atlases were specifically adapted or created for use within the Lead-DBS pipeline. The electric field (E-field) distribution was then simulated using an adaptation of the FieldTrip-SimBio pipeline that was integrated into Lead-DBS (https://www.mrt.uni-jena.de/simbio/; http://fieldtriptoolbox.org/) and thresholded at a level of 0.2 V/m[58].

For the London test cohort, we chose to use the original VTAs of the published study by Tyagi et al.[13]. These had instead been processed using Medtronic SureTune™ software and transferred into MNI space within the original study. The reason we chose to use the original VTAs were twofold. First, it would demonstrate generalizability of our findings (i.e. that our results could still be useful in case electrodes were localized using different software). Second, we aimed at yielding maximal transferability to the study by Tyagi et al.[13], which reported on the rich London dataset in more depth.

**Connectivity analysis**. Structural connectivity between VTAs and all other brain areas was calculated based on a normative connectome as similarly done in previous work[22,24,32,58,64]. Specifically, a whole-brain connectome based on state-of-the-art multi-shell diffusion-weighted imaging data from 985 subjects of the Human Connectome Project (HCP) 1200 subjects data release[52] was calculated in each patient using Lead-Connectome (www.lead-connectome.org). Whole-brain fiber tracts were normalized into standard space using a multispectral warp based on T1-weighted, T2-weighted, and diffusion-weighted acquisitions using ANTs (using the same "Effective Low Variance" preset implemented in Lead-DBS). In each subject, a total of 6000 fibers were sampled and aggregated to a joint dataset in standard space, resulting in a set of 6,000,000 fibers across 985 HCP subjects. For each of these tracts, a "Fiber T-score" was assigned by associating the fiber tract's connectivity to VTAs across patients with clinical outcome (Fig. 7). Specifically, (mass-univariate) two-sample _t_-tests between clinical outcomes in connected and unconnected VTAs were performed for all 6,000,000 tracts. Needless to say, these _T_-scores were not meant to result in significant results (given the mass-univariate nature of tests) but instead formed a model that could be used for out-of-sample predictions in other DBS cohorts. _T_-values from these tests should be seen as

"weights" and could be positive or negative (since two-sided _t_-tests were performed). A high absolute _T_-value meant that the fiber was strongly discriminative between good and poor responding VTAs or predictive for clinical outcome. For instance, a tract that was connected exclusively to VTAs in good responders (and not to VTAs of poor responders) would receive a high positive score. In return, a patient would most likely show more pronounced clinical benefit, if her/his VTA was strongly connected to many fibers with high positive _T_-values but not too many with negative scores. This analysis made it possible to assign aggregated fiber _T_-scores to each (out-of-sample) VTA in subsequent prediction analyses.

To account for the fact that larger VTAs would potentially automatically receive higher fiber _T_-scores, these were divided by the stimulation amplitude throughout the manuscript. Finally, Monte-Carlo random permutations (×1000) were conducted to obtain _p_-values, except for two-sample _t_-tests. This procedure is free from assumptions about the distributions (e.g. Student _t_ for _R_-values), which are typically violated in small sample sizes[65]. Scatterplots were visualized with 95% confidence bounds (gray or light-red areas).

**Reporting summary**. Further information on research design is available in the Nature Research Reporting Summary linked to this article.

## Data availability

The DBS MRI datasets generated during and analyzed during the current study are not publicly available due to data privacy regulations of patient data but are available from the corresponding author upon reasonable request. The resulting tract atlas is openly available within Lead-DBS software (www.lead-dbs.org).

## Code availability

All code used to analyze the dataset is openly available within Lead-DBS/-Connectome software (https://github.com/leaddbs/leaddbs).

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

## Acknowledgements

We would like to thank Kristy Kultas-Ilinsky and Igor Ilinsky, Rudolf Nieuwenhuys and Suzanne Haber for counseling regarding the anatomical validity of tractography results presented in this study. We would like to thank Hagai Bergmann, Helen Mayberg, Paul Krack, Christian Moll, Todd Herrington, Eduardo Alho and Erich Fonoff for very helpful general advice and feedback related to the present work. We would like to thank Cyril Pernet, Wolf-Julian Neumann and Roxanne Lofredi for statistical consultation and help on the manuscript. This work was supported by the German Research Foundation (Deutsche Forschungsgemeinschaft, Emmy Noether Stipend 410169619 to A.H., SPP 0141 to A.A.K. and KFO 219 to J.K.). The study was further funded by the Deutsche Forschungsgemeinschaft (DFG, German Research Foundation) – Project-ID 424778381 – TRR 295. We acknowledge support from the German Research Foundation (DFG) and the Open Access Publication Fund of Charité – Universitätsmedizin Berlin. Data were provided in part by the Human Connectome Project, WU-Minn Consortium (principal investigators: David Van Essen and Kamil Ugurbil; 1U54MH091657) funded by the 16 NIH Institutes and Centers that support the NIH Blueprint for Neuroscience Research; and by the McDonnell Center for Systems Neuroscience at Washington University.

## Author contributions

N.L. and A.H. conceptualized the study. N.L. and A.H. developed the software pipeline used, analyzed data and wrote the manuscript. J.C.B. conceptualized the study, acquired patient data and revised the manuscript. S.T. performed literature analyses and wrote the manuscript. H.A. acquired and processed patient data and revised the manuscript. G.J.B.E., A.B., and A.M.L. processed and analyzed human connectome data and revised the manuscript. B.A.-F. processed patient data, conceptualized part of the study and revised the manuscript. A.K., B.S., J.A.B., L.Z., E.J., S.C., V.V.-V., M.P., J.K. and A.A.K. acquired patient data and revised the manuscript.

## Competing interests

A.M.L. is consultant for Medtronic, Abbott and Boston Scientific. S.C. is consultant for Medtronic, Boston Scientific and Zimmer Biomet. M.P. has received honoraria for lecturing from the Movement Disorder Society, Medtronic, research support from Boston Scientific. J.K. has received financial support for investigator-initiated trials from Medtronic. A.A.K. reports personal fees and non-financial support from Medtronic, personal fees from Boston Scientific, grants and personal fees from Abbott outside the submitted work. A.H. reports lecture fees for Medtronic and Boston Scientific. N.L., J.C.B., A.K., S.T., H.A., G.J.B.E., A.B., B.A.-F., B.S., J.A.B., L.Z., E.J. and V.V.-V. have nothing to disclose.
