## [Peer Review File · Nature Communications]

Reviewers' Comments:

Reviewer #1:

Remarks to the Author:

The premise of the study by Li et al. is interesting and important. Evaluating possible similarities in the different DBS targets for OCD has potential to improve understanding of the general therapy. However, while the Horn group has generated substantial research excitement around the concept of using a "group connectome" for DBS analyses, the issues with this methodology are numerous and dangerous.

- First and foremost, the streamlines in these datasets are nearly all anatomical false positives [e.g. Schilling et al., NeuroImage, 2019]. Have the authors actually looked at the individual streamlines and said to themselves these look anatomically reasonable? My experience with these datasets is that they definitely do not.

- Second, human scale DWI and tractography is inherently biased toward large-size, high-anisotropy pathways. In the case of this study, the internal capsule (IC) swamps everything, and without any anatomical priors guiding the streamline generation process, no matter where you place your seed you simply get sucked into the IC. Therefore, you are only mapping long-range connections that are simply artifacts of the tracting algorithm and cannot be validated against the underlying neuroanatomy. The Yeh et al. [NeuroImage, 2018] atlas at least attempts to overcome some of these "group connectome" shortcomings with the definition of anatomically realistic streamline representations. This OCD DBS study needs to be re-run with the Yeh connectome to evaluate the robustness of the currently stated similarities between STN and ALIC DBS.

- Third, the compression of "patient-specific" electrode locations into an atlas space was abandoned by most DBS neurosurgery researchers long ago as being simply too inaccurate for detailed analyses on what is being stimulated. Recent studies have highlighted the multi-millimeter errors associated with atlas registration methods in the deep brain [e.g. Kim et al., HBM, 2018]. Given that you are trying to simulate the "connectivity" of a ~2mm activation volume, does it sound reasonable to do so from an anatomical base that is off by ~2mm, plus the electrode contact localization errors of ~1mm in the original patient space?

- Fourth, the statistical issues of "regression to the mean" when working in a tractography atlas space, which are compounded by the strong bias of the connectome's over-representation of long-range large-size pathways. Even more suspect, most of the ALIC VTAs show 0 "Fiber-T-Scores" (Figure 2). What does that tell you about this method? How do you then get a 0.56 correlation coefficient from that analysis? I seriously question the statistics of this study in general. For example, I see absolutely no correlation in the results presented in Figure 3, but the authors insist their finding are significant. Even if their statistical methods are valid, given the substantial limitations of the DBS modeling methods used, a correlation coefficient of 0.37 certainly does not justify the conclusions reached.

- None of these issues are adequately addressed (or even acknowledged) in the methods, results, or discussion.

Even more troubling with this paper is the audacity of the authors that these highly questionable results are ready for "open source" prospective clinical application by any untrained/unknowing users in an academic software package that has no regulatory oversight. This is both foolish and dangerous.

Reviewer #2:

Remarks to the Author:

In this manuscript, Li et al. present an interesting approach to link common clinical effects of different deep brain stimulation (DBS) targets for obsessive-compulsive disorder (OCD) through tractography modelling. The authors have analyzed data of two patients groups from two different hospitals using different targets (i.e. the anterior limb of the internal capsule (ALIC) or the subthalamic nucleus (STN)), and report that these targets share a white matter connection to the prefrontal cortex. The results are novel, address an important question for clinical neuroscience and provide support for a hypothesis that has been around for some time. I therefore think these results will be of interest for neuroscience, psychiatry and neurosurgery. However, there are certain issues that need to be addressed to ensure that the results are robust and unbiased.

Major Comments

1) It gets the impression that parts of the analysis may be circular. If I understood figure 6 correctly, a two-sample t-test was performed to select fibers that are associated with clinical response, and then to correlate predictive fibers to response. However, the selection of fibers is not independent from testing the relationship with response, and this is all done within the same sample. The resulting correlation between t-values and treatment outcome in the same sample is therefore likely to be optimistic (figure 2).

2) The next step was to test a model derived from sample 1 using data from sample 2 and vice versa, in which case there is no possible circularity. However, the results only present 1 correlation, suggesting that "Results of this cross-prediction across DBS centers and targets is shown in figure 3, bottom right ($R = 0.37$; $p = 0.027$)" is performed across both cohorts, rather than for each cohort independently. What are those corresponding R and P values?

3) The T-scores and response shown in figures 2 and 3 do not look Gaussian distributed, so non-parametric tests seem more appropriate. And there are multiple patients with $T=0$, regardless of treatment response, while in the text it is stated that the T-score is the summation of all fibers crossing through each individual's volume of tissue activated (VTA). It is hard to believe that the T-scores of all fibers within multiple VTAs were summed to be exactly 0 and not negative. Please clarify.

4) It is not entirely clear from the text what is shown in figure 3. It looks like the combined fibers (shown in the top panel) are the overlap in predictive fibers from both targets (so fewer fibers than the separate targets' fibers). However, judging from the range in T-scores,

the combined fibers are an addition of predictive fibers from both targets. How many of the common fibers were positively and negatively predictive? It appears that there are far more commonly positive predictive fibers than negatively predictive fibers.

5) Related to the previous question: if there is no normalization for number of fibers, then affecting as many fibers as possible (with a large VTA, due to a high voltage) would lead to a high T-score and a positively predicted outcome. In other words, there is no cost-function to the increasing the voltage to obtain a VTA that is connected to more positively predictive fibers. However, it is known that increasing the stimulation parameters beyond a certain level causes side effects instead of further improvement of the main effect. Could the authors comment on this?

6) To ensure that differences in treatment response are not explained by possible "trivial" differences in anatomical location of the VTAs rather than their connection to tracts, is it possible to show the VTAs of responders and non-responders? For example, in the ALIC group, it is suggested by the trajectories of the blue fibers that the non-responders were mostly stimulated ventrally to the responders. It could be that co-stimulation of the accumbens ventral to the ALIC (as was shown figure 1) was associated with non-response. Showing the VTAs could possibly strengthen the point the authors make in the introduction of the tract itself being a target. While it is clear that it is not the aim of the manuscript to determine whether the tract is solely responsible for treatment response, the authors at least suggest that there is a relationship between aiming for the tract and treatment response.

7) Fibers were all selected by association with treatment response in both targets, and in that sense data-driven. Please verify the validity of the presented fiber tract, is there an anatomical description? It is known that there are a lots of false positive fiber tract reconstructions in tractography (see e.g., Maier-Hein et al., Nature Comm., 2017).

Minor comments

8) For figure 2 and 3, it is not explained what the gray bars around the lines mean.

9) The authors use "top responders" at certain instances of the manuscript where "good responders" may be more appropriate. Top responders implies grouping of patients like responders vs. non-responders, though I get the impression that only change scores were used.

10) Please provide a reference for the 2% life-time prevalence of OCD.

Reviewers' comments:

Reviewer #1 (Remarks to the Author):

The premise of the study by Li et al. is interesting and important. Evaluating possible similarities in the different DBS targets for OCD has potential to improve understanding of the general therapy. However, while the Horn group has generated substantial research excitement around the concept of using a “group connectome” for DBS analyses, the issues with this methodology are numerous and dangerous.

- First and foremost, the streamlines in these datasets are nearly all anatomical false positives [e.g. Schilling et al., NeuroImage, 2019].

Have the authors actually looked at the individual streamlines and said to themselves these look anatomically reasonable? My experience with these datasets is that they definitely do not.

We discussed the tract of question in depth with Suzanne Haber, Rudolf Nieuwenhuys, Kristy Kultas-Illinsky and Igor Illinsky which are accomplished neuroanatomists.

According to their opinion, the tract does exist and is part of the ALIC. Since it does seem to connect to the subthalamic nucleus (fig. S1) it may represent fibers traversing from ventral prefrontal cortex to the STN.

We are not sure what the reviewer means with „these datasets“. The datasets we used were created by our own group and are a result of extensive methodological developments. The idea was introduced in Horn et al. 2014 NeuroImage and was validated in a high N study within Horn et al. 2016 NeuroImage. We first applied normative connectomes to a clinical population in Horn et al. 2017 NeuroImage and Ewert et al. 2017 NeuroImage. In the latter, the subthalamic nucleus was parcellated in MNI space using both a Parkinsonian and healthy normative connectome and the result matched findings from the influential Haynes and Haber 2013 macaque tracer study. The current dataset was constructed by the team in Toronto using the same methodology (Lead Connectome pipeline) on the whole HCP sample of 985 subjects that was acquired on specialized MRI hardware.

We still agree that fundamental issues are attached to dMRI-based tractography. The current study delineates and focuses on a single tract that has been well described in dissection studies and traverses within the extensively studied anterior limb of the internal capsule (see below). Thus, in the present case, the limitations of dMRI may not apply to a full extent. We do not use the method to describe novel anatomy or to quantify/characterize anatomical structures.

The deterministic fiber tracking method in DSI Studio (which was used) achieved the highest (92%) valid connection (ID#3) among 96 methods submitted from 20 different research groups, examined by a recent open competition (Maier-Hein, Nature Communications, 2017). The average accuracy was 54%. It was applied to state-of-the-art multishell dMRI data that was acquired at one of the leading imaging centers world-wide (Washington University St. Louis).

We agree that the tract needs anatomical confirmation and added an additional supplemental figure in which compares it to results from dissection studies. In our opin-

ion and based on discussions with anatomists (see above), the tract is well known and has been extensively described, in the literature.

We have added a supplementary figure (fig. S2) that shows our tract in synopsis with dissection results from anatomy textbooks. We also added the following paragraphs:

“Anatomically, the tract is a subpart of the well-characterized ALIC that connects the prefrontal cortex with the subthalamic nucleus and MD nucleus of the thalamus^{36,37}. Anatomical validity of the tract was discussed with four anatomists (see acknowledgement section). In the motor domain, the “hyperdirect pathway”, i.e., a direct connection from the frontal cortex to the subthalamic nucleus, has been well established^{38,39}, functionally, but the STN is known to receive widespread and direct input from the prefrontal cortex as a whole³⁶. Thus, the specific bundle delineated here may represent direct connectivity between frontal cortex and STN. An additional branch traversing from prefrontal cortex to the mediodorsal nucleus of the thalamus received slightly lower T-scores and is not shown in 3D visualizations but well visible in figure 6. The main branch of the tract traverses within a specific portion of the ALIC and follows its main structural course.” – results, p. 11

We also could not agree more that there are substantial flaws to the method of dMRI based tractography and added the following paragraph to our manuscript:

“It has been shown that dMRI-based tractography reconstructs a very high proportion of false-positive fibers in recent open challenges^{50,51}. We aimed at reducing the risk of false positive tractography results in three ways. First, we used the tracking method that achieved the highest (92%) valid connection score among 96 methods submitted from 20 different research groups in a recent open competition⁷⁵. Second, we used highest quality multi-shell diffusion data⁶⁸ acquired on a high N (985 subjects) at a state-of-the-art imaging center (HCP data acquired at Washington University in St. Louis, see Acknowledgements). Third, we compared the tract results with anatomy text-books and discussed its validity with four anatomists (see Acknowledgements). The tract described in the present study matches results from dissection studies (fig. S3). However, the potential that the tract represents a false positive result may not be completely ruled out given the fundamental limitations of dMRI-based tractography^{50,51}.” - discussion, page 19

- Second, human scale DWI and tractography is inherently biased toward large-size, high-anisotropy pathways. In the case of this study, the internal capsule (IC) swamps everything, and without any anatomical priors guiding the streamline generation process, no matter where you place your seed you simply get sucked into the IC. Therefore, you are only mapping long-range connections that are simply artifacts of the tracting algorithm and cannot be validated against the underlying neuroanatomy. The Yeh et al. [NeuroImage, 2018] atlas at least attempts to overcome some of these “group connectome” shortcomings with the definition of anatomically realistic streamline representations. This OCD DBS study needs to be re-run with the Yeh connectome to evaluate the robustness of the currently stated similarities between STN and ALIC DBS.

We would like to thank the reviewer for this excellent suggestion and would have loved to realize it. Unfortunately, the Yeh et al. atlas is not an exhaustive collection of tracts in the brain and the ALIC is not included in the dataset. Based on personal

communication with Fang-Cheng Yeh, the addition of further tracts is planned for the future but currently, no such data is available. As mentioned in our response to the issue above, we have added a novel figure and went into discussions with experts in the field that felt comfortable to be mentioned by name in the manuscript regarding the validity of the tract.

- Third, the compression of “patient-specific” electrode locations into an atlas space was abandoned by most DBS neurosurgery researchers long ago as being simply too inaccurate for detailed analyses on what is being stimulated. Recent studies have highlighted the multi-millimeter errors associated with atlas registration methods in the deep brain [e.g. Kim et al., HBM, 2018]. Given that you are trying to simulate the “connectivity” of a ~2mm activation volume, does it sound reasonable to do so from an anatomical base that is off by ~2mm, plus the electrode contact localization errors of ~1mm in the original patient space?

We agree that working in MNI space may lead to inaccuracies. Authors in the Kim et al. study used a linear mono-contrast method for their segmentation and we are not surprised by the large errors. Our registration pipeline is fundamentally different from most “standard off-the-shelf” neuroimaging pipelines (such as the FSL FLIRT method applied by Kim et al.) in that it was specifically designed and validated for deep brain stimulation. Differences include i) multispectral warps using a high-resolution MNI template ii) a specific subcortical refinement step, iii) brain shift correction, iv) use of a high-definition template and v) a phantom-validated electrode localization method (for details see Horn & Li et al. 2018 NeuroImage). Results of the pipeline generate automated STN and GPi segmentations that are as precise as manual expert segmentations (when compared to inter-rater accuracies) in both low-resolution and high-quality datasets (Ewert et al. 2019 NeuroImage).

Of course, we still agree that these analyses are biased and errors do occur. We added the following section to our limitations section:

“Inaccuracies in lead localization may result from the approach of warping electrodes into common space as done here. To minimize this issue, we used a modern neuroimaging pipeline that has been scientifically validated in numerous studies and involves advanced concepts such as brainshift correction, multispectral normalization, subcortical refinement steps⁷⁰ and phantom-validated electrode localizations⁷¹.” – discussion, page 19

- Fourth, the statistical issues of “regression to the mean” when working in a tractography atlas space, which are compounded by the strong bias of the connectome’s over-representation of long-range large-size pathways. Even more suspect, most of the ALIC VTAs show 0 “Fiber-T-Scores” (Figure 2). What does that tell you about this method? How do you then get a 0.56 correlation coefficient from that analysis? I seriously question the statistics of this study in general. For example, I see absolutely no correlation in the results presented in Figure 3, but the authors insist their finding are significant. Even if their statistical methods are valid, given the substantial limitations of the DBS modeling methods used, a correlation coefficient of 0.37 certainly does not justify the conclusions reached.

This is an excellent point and we have experienced “regression to the mean” situations in similar projects, ourselves, too. Here, however, the near zero results of Fiber-T-Scores indeed show that we are not dealing with a “regression to the mean” situa-

tion. In fact, the mean is shown (figure 2, tracts in green) while a specific tract at the border received high T-scores (in the ALIC group, a tract on top of the electrode cloud, in the STN group one traversing through the center but representing a ventral part of the overall connectivity pattern).

If results of the present study are to be believed, in the ALIC cohort, most electrodes have been implanted too ventral. This matches the clinical experience of the psychiatric colleagues in Cologne (most patients have now been switched to the most dorsal contact).

We added the following section explaining the zeros in our correlation plots. When replicating correlations without VTAs that received near zero scores, results remained significant (Fig. S1).

“Crucially, some VTAs of the ALIC cohort entirely resided below the identified tract and thus received a Fiber T-score of (near) zero (also see blue example patient in figure 3, bottom right). The same holds true when either learning the tract on the STN (Fig. 3) or the ALIC cohort itself (Fig. 2). To further investigate this matter, two-sample t-tests between improvements of patients with near zero scores (Fiber T-scores below 50) and the remaining patients with VTAs covering the tract well (scores above 50) were calculated. This showed that electrodes that reached the tract well resulted in significantly better clinical improvement ($T = 6.5$ at $p < 10^{-5}$ when the tract was defined by the ALIC cohort and $T = 3.7$ at $p < 0.005$ when it was defined by the STN cohort).” – results, p. 9

Moreover, we agree that the power of the initial study was low given the scarcity of DBS data in OCD world-wide. We now included an additional test-set graciously supplied by our colleagues in London (recently published in Tyagi et al., *Biological Psychiatry* 2018). Using the tract-model learned on the Grenoble and Cologne datasets, outcome in the London data could be predicted. Furthermore, the same tract emerged when analyzing the London data on its own, albeit being more right-lateralized than in the other centers. This led to the addition of the following novel paragraph and figure.

“The tract was then used to predict outcome in a completely independent test-set of patients that underwent surgery in a third center (London; Figure 4, right). Again, VTAs of patients that strongly overlapped fibers with positive scores and avoided tracts with negative scores received high Fiber T-scores which significantly correlated with empirical improvement, across the group ($R = 0.75$ at $p = 0.040$).” – results, page 10

Figure 4. Predictive fibers when including all patients from both cohorts (Cologne, Grenoble) irrespective of their target (top). The sum of aggregated Fiber T-scores under each VTA explained % Y-BOCS improvement (bottom left). Moreover, the tract defined in one target could cross-predict improvement in the cohort treated with the other target (bottom right). Red fibers are positively associated with clinical improvement, blue fibers negatively.

- None of these issues are adequately addressed (or even acknowledged) in the methods, results, or discussion.

We would like to apologize and have extensively worked on our limitations section. The following additional paragraphs are the most relevant:

“Second, we used normative connectome data instead of patient-specific diffusion-weighted MRI data (which is unavailable for most of the patients included). This poses dramatic limitations given that such data cannot be representative of patient-specific anatomical variations. Still, we argue that some aspects about general pathophysiological mechanisms can still be investigated using normative data.” – discussion, page 18

“Inaccuracies in lead localization may result from the approach of warping electrodes into common space as done here. To minimize this issue, we used a modern neuroimaging pipeline that has been scientifically validated in numerous studies and involves advanced concepts such as brain shift correction, multispectral normalization, subcortical refinement steps⁷³ and phantom-validated electrode localizations⁷⁴.” – discussion, page 19

“Importantly, given the correlative nature of the study, our findings may not differentiate between local and global effects. For instance, the tracts may have spuriously originated in the ALIC group because a more dorsal stimulation resulted with better clinical outcome. The congruency between results of the STN- and ALIC-cohorts resulting in the same fiber bundle may still suggest that the identified tract could play a causal role. However, such a claim would need to be confirmed e.g. using optogenetics in animal studies.” – discussion, page 19

“It has been shown that dMRI-based tractography reconstructs a very high proportion of false-positive fibers in recent open challenges^{50,51}. We aimed at reducing the risk of false positive tractography results in three ways. First, we used the tracking meth-

od that achieved the highest (92%) valid connection score among 96 methods submitted from 20 different research groups in a recent open competition⁵⁰. Second, we used highest quality multi-shell diffusion data⁷¹ acquired on a high N (985 subjects) at a state-of-the-art imaging center (HCP data acquired at Washington University in St. Louis, see Acknowledgements). Third, we compared the tract results with anatomy text-books and discussed its validity with four anatomists (see Acknowledgements). The tract described in the present study matches results from dissection studies (fig. S3). However, the potential that the tract represents a false positive result may not be completely ruled out given the fundamental limitations of dMRI-based tractography^{50,51}.” – discussion, page 19

Even more troubling with this paper is the audacity of the authors that these highly questionable results are ready for “open source” prospective clinical application by any untrained/unknowing users in an academic software package that has no regulatory oversight. This is both foolish and dangerous.

We would like to remain on our trajectory to propagate the most transparent and open science possible. This includes openly releasing datasets resulting from research articles. We still agree this could become dangerous if colleagues would apply datasets blindly and added an additional note that the atlas should not be used for clinical decision making (and if at all only within research studies with rigorous IRB approval).

“Of note, Lead-DBS is scientific and not clinical software and the tract should not be used for clinical decision making⁵².” – discussion, p. 11

“We must emphasize that this study presents group data and utilizes connectivity from individuals without OCD. As mentioned by others in this very context, we could not agree more that surgical decision making for DBS should not be based on aggregated normative data alone⁵². Further studies are required to determine whether individual patient connectivity or generic connectome data can assist with optimization of surgical targeting by determining the crossing sites of symptom networks for each specific patient.” – discussion, p. 18

Reviewer #2 (Remarks to the Author):

In this manuscript, Li et al. present an interesting approach to link common clinical effects of different deep brain stimulation (DBS) targets for obsessive-compulsive disorder (OCD) through tractography modelling. The authors have analyzed data of two patients groups from two different hospitals using different targets (i.e. the anterior limb of the internal capsule (ALIC) or the subthalamic nucleus (STN)), and report that these targets share a white matter connection to the prefrontal cortex. The results are novel, address an important question for clinical neuroscience and provide support for a hypothesis that has been around for some time. I therefore think these results will be of interest for neuroscience, psychiatry and neurosurgery. However, there are certain issues that need to be addressed to ensure that the results are robust and unbiased.

We would like to thank the reviewer for the positive overall evaluation of our work.

Major Comments

- 1) I get the impression that parts of the analysis may be circular. If I understood figure 6 correctly, a two-sample t-test was performed to select fibers that are associated with clinical response, and then to correlate predictive fibers to response. However, the selection of fibers is not independent from testing the relationship with response, and this is all done within the same sample. The resulting correlation between t-values and treatment outcome in the same sample is therefore likely to be optimistic (figure 2).

We now report cross-predictions individually which were significant for both directions. We also made certain to use the verbs “explain” and “predict” accurately throughout the manuscript to differentiate within- and out-of-sample results. A few sections were added, the following being the most central to the topic:

“Of note, these correlations are somewhat circular and meant to describe the degree of how well the tracts could explain the sample of patients on which they had been built. More interestingly, in the next step, the tract learned exclusively on the STN cohort was used to explain outcome in the ALIC cohort and vice versa (figure 3). Here, the STN-based tract could significantly predict outcome in the ALIC cohort ($R = 0.50$ at $p = 0.009$) and vice-versa ($R = 0.49$ at $p = 0.041$).” – results, page 8

- 2) The next step was to test a model derived from sample 1 using data from sample 2 and vice versa, in which case there is no possible circularity. However, the results only present 1 correlation, suggesting that “Results of this cross-prediction across DBS centers and targets is shown in figure 3, bottom right ($R = 0.37$; $p = 0.027$)” is performed across both cohorts, rather than for each cohort independently. What are those corresponding R and P values?

We agree – as mentioned in the last answer, we now report cross-predictions separately (see above). We also added a third test-set from the London DBS centers and replicated main results on this additional dataset:

“The tract was then used to predict outcome in a completely independent test-set of patients that underwent surgery in a third center (London; Figure 4, right). Again, VTAs of patients that strongly overlapped fibers with positive scores and avoided tracts with negative scores received high Fiber T-scores which significantly correlated with empirical improvement, across the group ($R = 0.75$ at $p = 0.040$).” – results, page 10

- 3) The T-scores and response shown in figures 2 and 3 do not look Gaussian distributed, so non-parametric tests seem more appropriate. And there are multiple patients with $T=0$, regardless of treatment response, while in the text it is stated that the T-score is the summation of all fibers crossing through each individual's volume of tissue activated (VTA). It is hard to believe that the T-scores of all fibers within multiple VTAs were summed to be exactly 0 and not negative. Please clarify.

We now used non-parametric tests throughout the manuscript and added this to the methods section:

“Random permutation ($\times 5000$) was conducted to obtain p-values, except for two-sample T-Tests.” – methods, p. 23

As mentioned in our response to reviewer 1, we agree that the near zero-scores may seem irritating at first but they do in fact tell us a crucial property about some VTAs in

the ALIC cohort. Specifically, those VTAs resided largely below the tract, thus receiving a low score. We added illustrative examples to novel figure 3 that will hopefully lead to a better understanding of how the low scores originate. Correlations of the ALIC cohort remain significant when leaving out patients with VTAs that resided outside of the tract (figure S1).

We added the following section explaining the zeros in our correlation plots:

“Crucially, some VTAs of the ALIC cohort entirely resided below the identified tract and thus received a Fiber T-score of (near) zero (also see blue example patient in figure 3, bottom right). The same holds true when either learning the tract on the STN (Fig. 3) or the ALIC cohort itself (Fig. 2). To further investigate this matter, two-sample *t*-tests between improvements of patients with near zero scores (Fiber T-scores below 50) and the remaining patients with VTAs covering the tract well (scores above 50) were calculated. This showed that electrodes that reached the tract well resulted in significantly better clinical improvement ($T = 6.5$ at $p < 10^{-5}$ when the tract was defined by the ALIC cohort and $T = 3.7$ at $p < 0.005$ when it was defined by the STN cohort).” – results, p. 9

“Figure 3. Cross-prediction between the Cologne / ALIC and Grenoble / STN cohorts. Top: The tract exclusively learned on the ALIC cohort was used to calculate Fiber T-scores for all patients in the STN cohort. One example patient with strong overlap of the tract (yellow) received a high Fiber T-score, whereas one with less overlap received a lower score (blue). The two example patients are marked in the correlation plot on the left. Bottom: Here, the tract was learned on the STN cohort to predict the ALIC patients. Again, two example patients are shown. Of note, here, some VTAs barely overlapped with the tract and consequently received a near-zero score.”

4) It is not entirely clear from the text what is shown in figure 3. It looks like the combined fibers (shown in the top panel) are the overlap in predictive fibers from both targets (so fewer fibers than the separate targets' fibers). However, judging from the range in T-scores, the combined fibers are an addition of predictive fibers from both targets. How many of the common fibers were positively and negatively predictive? It appears that there are far more commonly positive predictive fibers than negatively predictive fibers.

We agree that results from the original figure 3 were a bit misleading, and, as suggested by the reviewer, opted to show the cross-prediction results separately for each cohort (i.e. STN -> ALIC and ALIC -> STN) in novel figure 3 (see above). We now also report the ratio between positive and negative fibers which is different for the two targets:

“Depending on the target, the analysis revealed different proportions of “positive” and “negative” fibers (ALIC cohort: 22.2k positive vs. 1.9k negative fibertracts selected from the groupconnectome; STN cohort: 45.1k positive vs. 48.6k negative fibers and both cohorts combined: 54.4k positive vs. 9.6k negative fibers).” – results, p. 10

5) Related to the previous question: if there is no normalization for number of fibers, then affecting as many fibers as possible (with a large VTA, due to a high voltage) would lead to a high T-score and a positively predicted outcome. In other words, there is no cost-function to the increasing the voltage to obtain a VTA that is connected to more positively predictive fibers. However, it is known that increasing the stimulation parameters beyond a certain level causes side effects instead of further improvement of the main effect. Could the authors comment on this?

We would like to thank the reviewer for this excellent remark. We now divide fiber T-scores by stimulation amplitude throughout the manuscript and this improved every single result of the manuscript. The following sentence was added to the methods section:

“To account for the fact that larger VTAs would potentially automatically receive higher T-scores, reported Fiber T-scores were divided by the stimulation amplitude throughout the manuscript.” – methods, p. 23

6) To ensure that differences in treatment response are not explained by possible “trivial” differences in anatomical location of the VTAs rather than their connection to tracts, is it possible to show the VTAs of responders and non-responders? For example, in the ALIC group, it is suggested by the trajectories of the blue fibers that the non-responders were mostly stimulated ventrally to the responders. It could be that co-stimulation of the accum-bens ventral to the ALIC (as was shown figure 1) was associated with non-response. Showing the VTAs could possibly strengthen the point the authors make in the introduction of the tract itself being a target. While it is clear that it is not the aim of the manuscript to determine whether the tract is solely responsible for treatment response, the authors at least suggest that there is a relationship between aiming for the tract and treatment response.

We agree that it is impossible to disentangle local from global effects using our methods and added the following paragraph to the limitations section of our manuscript:

“Importantly, given the correlative nature of the study, our findings may not differentiate between local and global effects. For instance, the tracts may have spuriously originated in the ALIC group because a more dorsal stimulation resulted with better clinical outcome. The congruency between results of the STN- and ALIC-cohorts resulting in the same fiber bundle may still suggest that the identified tract could play a causal role. However, such a claim would need to be confirmed e.g. using optogenetics in animal studies.” – discussion p. 19.

Furthermore, we do agree that ventral stimulation in the ALIC group was associated with poor response – and we added novel figure 3 (see above) that shows four patient examples of responders and non-responders.

7) Fibers were all selected by association with treatment response in both targets, and in that sense data-driven. Please verify the validity of the presented fiber tract, is there an anatomical description? It is known that there are a lots of false positive fiber tract reconstructions in tractography (see e.g., Maier-Hein et al., Nature Comm., 2017).

We strongly agree that the tract needs to be anatomically valid to be meaningful. As mentioned in responses to reviewer 1, we had discussed the validity of the tract with three anatomists and have now added a supplementary figure that compares the tract with results from dissection studies (figure S2).

Furthermore, the following sections were added:

“We aimed at reducing the risk of false positive tractography results in three ways. First, we used the tracking method that achieved the highest (92%) valid connection score among 96 methods submitted from 20 different research groups in a recent open competition⁵⁰. Second, we used highest quality multi-shell diffusion data⁷¹ acquired on a high N (985 subjects) at a state-of-the-art imaging center (HCP data acquired at Washington University in St. Louis, see Acknowledgements). Third, we compared the tract results with anatomy text-books and discussed its validity with four anatomists (see Acknowledgements). The tract described in the present study matches results from dissection studies (fig. S3). However, the potential that the tract represents a false positive result may not be completely ruled out given the fundamental limitations of dMRI-based tractography^{50,51}.” – discussion, page 19

“Anatomically, the tract is a subpart of the well-characterized ALIC that connects the prefrontal cortex with the subthalamic nucleus and MD nucleus of the thalamus^{36,37}. Anatomical validity of the tract was discussed with four anatomists (see acknowledgement section). In the motor domain, the “hyperdirect pathway”, i.e., a direct connection from the frontal cortex to the subthalamic nucleus, has been well established^{38,39}, functionally, but the STN is known to receive widespread and direct input from the prefrontal cortex as a whole³⁶. Thus, the specific bundle delineated here may represent direct connectivity between frontal cortex and STN. An additional branch traversing from prefrontal cortex to the mediodorsal nucleus of the thalamus received slightly lower T-scores and is not shown in 3D visualizations but well visible in figure 6. The main branch of the tract traverses within a specific portion of the ALIC and follows its main structural course.” – results, p. 11

Minor comments

8) For figure 2 and 3, it is not explained what the gray bars around the lines mean.

We agree this should be specified:

“Scatterplots in the present manuscript are visualized with 95% confidence bounds (gray or light-red areas).” – methods, p. 23

9) The authors use “top responders” at certain instances of the manuscript where “good responders” may be more appropriate. Top responders implies grouping of patients like responders vs. non-responders, though I get the impression that only change scores were used.

We agree and changed this to good responders.

10) Please provide a reference for the 2% life-time prevalence of OCD.

We added a reference (Ruscio et al., 2010).

Reviewers' Comments:

Reviewer #2:

Remarks to the Author:

The authors have addressed all my concerns. The results seem robust by using nonparametric statistics and testing the model across sites. The authors now even included a third site for further validation. And the additional figures provide additional evidence that the discovered tracts are anatomically plausible.

I do share the reservation of the other reviewer that group tracts are not suitable for surgical planning for individual patients. But the method appears justified for the current aim to determine the overlap between tracts of different stimulation targets.

Reviewer #3:

Remarks to the Author:

Li et al. present an interesting approach of using normative high quality data from the human connectomics project to evaluate and predict subject individual outcome in DBS treated patients with OCD.

While the idea of using diffusion tractography based information to stimulate afferent and efferent networks with DBS rather than multi-circuitry connection centres like STN or ALIC is certainly not new, the hereby-presented work is able to stimulate and inspire more people working in this field and provide feedback on larger sample sizes.

I have no general objection with the study itself, its scientific hypothesis and the way the results are analysed and presented. Due to its limited sample size, it can only be rewarded as a preliminary study with less than statistically solid results but the authors themselves do state this as one major limitation. Therefore, I am not sure it is well suited for an article in Nature Communications.

In my view, it is clear that even though the data are made publically available and the analysis software is open source, the approach should not be followed vacuously and irresponsible to clinically treat individual patients. This is a scientific paper without proposing a clinically approved treatment procedure, which the authors make very clear on various text passages.

As a general statement, I wish the authors would more clearly acknowledge the work of Coenen et al. from 2009-2013, pioneering the idea of stimulating or modulating brain networks instead of commonly used neurosurgical target areas like brain nuclei (which they further down cite as a leading example "for the case of treatment-refractory depression 20". From:

63: "In parallel, DBS has experienced a conceptual paradigm-shift away from focal stimulation of specific brain nuclei (such as the subthalamic nucleus or globus pallidus in Parkinson's Disease; PD) toward modulating distributed brain networks (such as the motor basal ganglia cortical cerebellar loop in PD) 17-19."

Here the authors only cite their own work giving the impression that this “paradigm shift” roots in their own ingenuity.

Moreover, I have some questions and concerns in regards to the integrity and consistency of the various source data and their conjunction.

Total NOP: 22+14 (36), London group: 4 electrodes but 6 patients ? N=6 -> total 42

121: Only as a final validation step, an additional test-cohort was included (London; four electrodes targeting both ALIC and STN).

But these were 6 patients. So four electrodes per patients or what does the “four electrodes mean? Please specify!

154: The degree of lead connectivity to this tract correlated with clinical improvement ($R = 0.63$ at $p < 0.001$ in the ALIC cohort and $R = 0.77$ 156 at $p < 0.001$ in the STN cohort; Figure 2, bottom row).

That is quite a low statistical correlation with an R^2 of below 0.4 for ALIC. One could also argue that with such low confidence there does not exist a statistical significant correlation. On what test and which variable (Linear regression with variable slope?) are the p-values based?

The same fundamental question applies to the regression plots in fig.3.

246: Figure 4. Predictive fibers when including all patients from both cohorts (Cologne, Grenoble) irrespective of their target (top). The sum of aggregated Fiber T-scores under each VTA explained % YBOCS improvement (bottom left). Moreover, the tract defined in one target could cross-predict improvement in the cohort treated with the other target (bottom right). Red fibers are positively associated with clinical improvement, blue fibers negatively.

Please extend the figure caption. There are only two figures (left and right) in the manuscript. What do you refer to when you mention: top, bottom left, bottom right?

The linear regression model shown on Fig4-left:

I can only deduct 6 data points (are several others overlapping?). From these data points I will get a regression analysis of $R=0.75$ and a significance level of 0.09 (linear regression ANOVA).

That will result in a $R^2=0.55$ and an adjusted $R^2=0.44$ only with a p-value for the slope of 0.09.

This does not really constitute a finding that I would describe as a “prediction between Y-BOCS improvement” based on the fiber T-score. Aren’t you exaggerating a hypothesis based on too few solid data points? Please comment.

372: “Data of such quality can usually not be acquired in individual patients

and test-retest scan reliability in DBS settings has shown to be poor even when applying state-of-the-art MRI protocols".

That statement is highly disputable. As far as I am aware, it has not been proven yet what the better concept might be: complete patient individual data sets (MRI, DWI with high quality fibertracking ability (multi-tensor, multi-shell), CT and VAT simulation) or normative approaches like using brain atlases, MRI templates, human connectome data from a different cohort etc.

In my point of view, this paper could non-arguably start such a discussion and promote as well as stimulate further studies and as such, I would like to welcome and invite the authors for further investigation in this field.

434: For all patients in the three cohorts, high-resolution structural T1-weighted images were acquired on a 3.0-Tesla MRI-scanner, before surgery. Postoperative computer tomography (CT) was obtained in twenty-five patients after surgery to verify correct electrode placement, while eleven patients from the Grenoble cohort and the six London patients received postoperative MRI instead.

This constitutes a problem: from a total of 42 patients only 25 received a post-op CT to confirm accurate location of the DBS electrodes and derive precise position of the individual DBS leads. The rest (17) received post-op MRI's. How were those data obtained? At 1.5T or at 3T? 3T is not clinically approved for DBS post-op DBS assessment while 1.5T MRI has relatively poor SNR to well define the target structures and provide high SNR data for good registration to the MNI templates.

Additionally how well could you define the DBS lead points based on MRI images that are highly distorted in the vicinity of the electrodes. In my point of view this constitute a major flaw in the study that will lead to inconsistencies and congruent data processing outcome.

Reviewers' comments:

Reviewer #2 (Remarks to the Author):

The authors have addressed all my concerns. The results seem robust by using nonparametric statistics and testing the model across sites. The authors now even included a third site for further validation. And the additional figures provide additional evidence that the discovered tracts are anatomically plausible.

I do share the reservation of the other reviewer that group tracts are not suitable for surgical planning for individual patients. But the method appears justified for the current aim to determine the overlap between tracts of different stimulation targets.

We would like to thank the reviewer for the positive evaluation and helpful comments that definitely improved our manuscript at large.

Reviewer #3 (Remarks to the Author):

Li et al. present an interesting approach of using normative high quality data from the human connectomics project to evaluate and predict subject individual outcome in DBS treated patients with OCD.

While the idea of using diffusion tractography based information to stimulate afferent and efferent networks with DBS rather than multi-circuitry connection centres like STN or ALIC is certainly not new, the hereby-presented work is able to stimulate and inspire more people working in this field and provide feedback on larger sample sizes.

I have no general objection with the study itself, its scientific hypothesis and the way the results are analysed and presented. Due to its limited sample size, it can only be rewarded as a preliminary study with less than statistically solid results but the authors themselves do state this as one major limitation. Therefore, I am not sure it is well suited for an article in Nature Communications.

We agree that the sample size is limited although the Cologne cohort alone (N = 22) is one of the largest OCD-DBS cohorts worldwide and together, an N = 42 patients could be considered a large sample. We now included a fourth cohort (see below), leading up to 50 patients in total. We tried our best to critically evaluate and validate our results by cross-validations across the four centers but agree that results need to be further validated, in the future. In fact, several groups world-wide have now taken first steps to prospectively validate our results in upcoming clinical trials. An OCD-DBS database is planned for the future (spearheaded by Suzanne Haber and Robert Malenka) and we are in contact with this endeavor and very curious to see results to become prospectively validated.

We agree that treating circuits instead of grey matter nodes is not new. For instance, Talairach and Leksell began lesioning the anterior limb of the internal capsule in patients with OCD (and other psychiatric diseases), already with the goal of disrupting limbic input to the prefrontal cortex as early as 1950. Even before that, Otfried Foerster or Oskar & Helene Vogt analyzed impact of tracts on disorders of the basal ganglia as early as 1921.

What is new in the current study is that we show a clear relationship between clinical outcome and the degree of modulating a specific sub-bundle of the anterior limb of the internal capsule. We do so in a data-driven sense and while we “learn” the specific bundle based on data in one target (e.g. ALIC), we explain a significant part of the variance in patients from the other target (STN and vice versa). We show that “learning” the tract on either target results in the same anatomical structure. To the best of our knowledge, such a relationship (able to explain variance in out-of-sample data) has never been shown in DBS for OCD. Furthermore, the concept of learning a tract-target at one site and explaining variance in a different site has never been achieved in DBS for any disease.

For instance, the often-reproduced and seminal paper by Schlaepfer et al. 2014 in Neuropsychopharmacology shows that most DBS sites used to treat depression are anatomically connected to a white matter tract, it does not show empirical evidence that modulating the tract is indeed functionally relevant for clinical outcome.

In other words, while numerous studies have shown that DBS targets are anatomically connected (e.g. by the Coenen or Mayberg groups), it has not been shown that modulating these connections at different sites of the tract is indeed predictive of clinical improvement. Even in the seminal “depression switch” papers by the Mayberg group, such a relationship has not been shown in out-of-sample data, let alone across cohorts, DBS centers and DBS targets.

We do believe that this may indeed make the present article a significant to the field of OCD-DBS but also to neuromodulation at large.

We have added the following paragraph to address this:

“While the concept of modulating white-matter tracts (instead of grey matter nuclei) is certainly not new (and anterior capsulotomy was introduced already in the ~1950ies by Talairach and Leksell (Feldman and Goodrich, 2001)²⁴), novel MRI technologies such as diffusion-weighted imaging based tractography have been incorporated to functional neurosurgery in order to more deliberately target white-matter tracts (Henderson, 2012)²¹. In this translational development, the Coenen and Mayberg groups should be explicitly mentioned, among others, for pioneering and rapidly translating the use of tractography to functional surgery since around 2009 (Coenen et al., 2017; Coenen et al., 2011; Choi et al., 2015; Riva-Posse et al., 2018; Coenen et al., 2011; Riva-Posse et al., 2014)^{13,18–20,25,26}.” – introduction, p. 3

In my view, it is clear that even though the data are made publically available and the analysis software is open source, the approach should not be followed vacuously and irresponsibly to clinically treat individual patients. This is a scientific paper without proposing a clinically approved treatment procedure, which the authors make very clear on various text passages.

We could not agree more to this and as mentioned by the reviewer have aimed at making this very clear in many passages of the manuscript. We do believe, however, that functional neurosurgeons world-wide will not follow such concepts vacuously. Such changes in established routines – if they should happen at all – should be slow, care- and thoughtful processes that evolve in tiny steps. We plan at least three more methodological papers before thinking about suggesting to translating any of the presented work into clinical practice.

This being said, while DBS is established for movement disorders, it is not as much for OCD and colleagues world-wide seemed to be very curious and thankful about any hint that could help gain a clearer picture of effective stimulation sites, so far. Some of our insights might be translatable without changing any of the routine.

For instance, clinical experience in the Cologne cohort was that the dorsal-most contacts in their ALIC cohort had best effects. Most patients in this cohort are now programmed on the most dorsal contact. This experience was made parallel and before our studies but matches their results. This kind of knowledge could be easier and safer to translate into clinical practice before changing any of the surgical routines or targets.

As a general statement, I wish the authors would more clearly acknowledge the work of Coenen et al. from 2009-2013, pioneering the idea of stimulating or modulating brain networks instead of commonly used neurosurgical target areas like brain nuclei (which they further down cite as a leading example “for the case of treatment-refractory depression 20”).

From:

63: *“In parallel, DBS has experienced a conceptual paradigm-shift away from focal stimulation of specific brain nuclei (such as the subthalamic nucleus or globus pallidus in Parkinson’s Disease; PD) toward modulating distributed brain networks (such as the motor basal ganglia cortical cerebellar loop in PD) 17–19.”*

Here the authors only cite their own work giving the impression that this “paradigm shift” roots in their own ingenuity.

We are very sorry for this oversight and it was not our plan to give this impression. We are well aware of the many pioneering works that have been published in this field. While we had already dedicated the whole second paragraph of our manuscript to the pioneering work by the Coenen group, we have now changed the specific sentence and reference works from the Coenen group and various other groups (and not our own work) in the following:

“In parallel, DBS has experienced a conceptual paradigm-shift away from focal stimulation of specific brain nuclei (such as the subthalamic nucleus or globus pallidus in Parkinson’s Disease; PD) toward modulating distributed brain networks (such as the motor basal ganglia cortical cerebellar loop in PD) (Coenen et al., 2017; Reinacher et al., 2018; Coenen et al., 2011; Choi et al., 2015; Riva-Posse et al., 2018; Henderson et al., 2012; Petersen et al., 2019; Gunalan et al., 2001)^{13,17-23} – introduction, p. 3

We furthermore added the following paragraph to the introduction to further acknowledge work by the Coenen group as (also see first issue):

“While the concept of modulating white-matter tracts (instead of grey matter nuclei) is certainly not new (and anterior capsulotomy was introduced already in the ~1950ies by Talairach and Leksell (Feldman and Goodrich, 2001)²⁴), novel MRI technologies such as diffusion-weighted imaging based tractography have been incorporated to functional neurosurgery in order to more deliberately target white-matter tracts (Henderson, 2012)²¹. In this translational development, the Coenen and Mayberg groups should be explicitly mentioned, among others, for pioneering and rapidly translating the use of tractography to functional surgery since around 2009 (Coenen et al., 2018;

Coenen et al., 2011; Choi et al., 2015; Riva-Posse et al., 2018; Coenen et al., 2011; Riva-Posse et al., 2014)^{13,18-20,25,26}. – introduction, p. 3

Moreover, I have some questions and concerns in regards to the integrity and consistency of the various source data and their conjunction.

Total NOP: 22+14 (36), London group: 4 electrodes but 6 patients ? N=6 -> total 42

121: *Only as a final validation step, an additional test-cohort was included (London; four electrodes targeting both ALIC and STN).*

But these were 6 patients. So four electrodes per patients or what does the “four electrodes mean? Please specify!

We apologize that this remained unclear. The 22 + 14 patients from Cologne and Grenoble had two electrodes each (N = 36 patients with N = 72 electrodes), the 6 patients in the London group had four electrodes each (N = 6 patients with N = 24 electrodes).

A fourth center has now been added (Madrid), where 8 patients received 16 electrodes.

We added the following sentence to further clarify this:

“The patients from Cologne, Grenoble and Madrid received two electrodes each (N = 44 patients with N = 88 electrodes), the six patients in the London cohort received four electrodes each (N = 6 patients with N = 24 electrodes).” – methods, p. 23,

We also added a row in table 1 for electrodes and thus specified numbers of patients and electrodes separately for each cohort.

154: *The degree of lead connectivity to this tract correlated with clinical improvement (R = 0.63 at $p < 0.001$ in the ALIC cohort and R = 0.77 156 at $p < 0.001$ in the STN cohort; Figure 2, bottom row).*

That is quite a low statistical correlation with an R2 of below 0.4 for ALIC. One could also argue that with such low confidence there does not exist a statistical significant correlation.

On what test and which variable (Linear regression with variable slope?) are the p-values based?

The same fundamental question applies to the regression plots in fig.3.

An R2 of ~0.4 would explain ~40% of variance in clinical improvements. Again, we would like to emphasize that these are out-of-sample predictions, i.e. likely not inflated by overfitting. We agree with the reviewer that explaining even more variance would be optimal, e.g. to be able to predict outcome in individual patients. In our view, since we are using electrode placement (and derived connectivity) as the only regressor in our model, its explanatory value could also be considered high (it is much higher than in Parkinson’s Disease, where the highest amount of variance explained has usually been around 20% but results are consistent across three studies of different centers, see e.g. Horn 2019 Curr Op Neurol for an overview).

Still, we need to ask ourselves where the noise term may derive, i.e. where the unexplained proportion of variance (~60%) comes from. First, our independent variable is a clinical score which has a low test-retest reliability. We do think this adds a large amount of noise to the model. Patients may have a good or poor day while the exam

is taken, different observers may judge symptoms slightly differently. To this end, our data from all four cohorts was derived exclusively from clinical studies so was taken under the best conditions possible at each center.

The second big factor may be imprecision of electrode localizations & modeling. We did our best to maintain the highest standards of precision here, but this is a clear limitation that we address at length in the manuscript. Finally, noise could be introduced by patient-inhomogeneity and patient-specific anatomical variations of the tract that we isolated. All these factors should bias our results toward non-significance. On the contrary, explaining ~40% of the variance by a single imaging-based regressor in out-of-sample correlations could in our view be considered an important finding.

As specified, *p*-values were obtained by conducting non-parametric Monte Carlo permutation tests. The advantage of these resampling based methods is that they do not depend on an assumption made on the distribution of the statistical metric (e.g. Student T distribution for *R*). This assumption is typically violated in small samples as present here (e.g. see Good et al. 2005 *Permutation, Parametric, and Bootstrap Tests of Hypotheses*, Springer). Instead, using permutation-tests for correlation estimates is considered more robust and is widely used in the neuroscientific literature to control the false discovery rate (e.g. Maris and Oostenveld, 2007, *Journal of Neuroscience Methods*; Narayanan and Laubach, 2009, *Methods in Molecular Biology*; Tian et al, 2016, *Neuro Image*; Kim et al., 2019, *The Journal of Neuroscience*).

Specifically, the degrees of lead connectivity to the predictive tracts (“*T*-scores”) were randomly shuffled for 1000 times to determine the original correlation coefficient (two-tailed Pearson’s *R*) within the distribution of surrogates. The *p*-values were calculated as the proportion of *R* values that were at least as large as the original *R*, which was derived from non-permuted data. (see https://www.uvm.edu/~statdhtx/StatPages/Randomization%20Tests/RandomCorr/randomization_Correlation.html) for a general primer.

We further clarified this in the methods section:

“Monte-Carlo random permutations ($\times 1000$) were conducted to obtain *p*-values, except for two-sample *t*-tests. This procedure is free from assumptions about the distributions (e.g. Student-*T* for *R*-values) which are typically violated in small sample sizes (Good, 2005)⁹¹.” – methods, p. 25

246: Figure 4. Predictive fibers when including all patients from both cohorts (Cologne, Grenoble) irrespective of their target (top). The sum of aggregated Fiber *T*-scores under each VTA explained % YBOCS improvement (bottom left). Moreover, the tract defined in one target could cross-predict improvement in the cohort treated with the other target (bottom right). Red fibers are positively associated with clinical improvement, blue fibers negatively.

Please extend the figure caption. There are only two figures (left and right) in the manuscript. What do you refer to when you mention: top, bottom left, bottom right?

The linear regression model shown on Fig4-left:

I can only deduct 6 data points (are several others overlapping?). From these data points I will get a regression analysis of $R=0.75$ and a significance level of 0.09 (linear regression ANOVA).

That will result in a $R^2=0.55$ and an adjusted $R^2=0.44$ only with a p-value for the slope of 0.09.

This does not really constitute a finding that I would describe as a “prediction between Y-BOCS improvement” based on the fiber T-score. Aren’t you exaggerating a hypothesis based on too few solid data points? Please comment.

We are sorry for the oversight, we now corrected the figure caption (see below). In this figure, we performed a Pearson-Correlation with random permutations to obtain p-values. This has been standard in our lab for a while and is widely used when dealing with small sample sizes (see above). However, in this case, the classic two-sided R-to-p based p-value would indeed be 0.089. Given the confirmatory nature of this very example (confirming results from Grenoble/Cologne based on London data), a one-sided test would be in order which ranks at $p = 0.044$.

Still, we agree that this should be made very clear & transparent and we now explicitly state the permutation-based nature of the test in the figure caption and report the two-/one-sided classic test results (see below). We do agree (and specify this in the limitations section) that the data from the London group should be interpreted cautiously given the low N of 6 patients. However, as mentioned, we now have added data from a fourth cohort which again was completely naïve to our model which significantly predicted variance in the outcome in these patients, as well (N = 8 patients tested at N = 4 time points each resulting in 32 sample points). See novel figure 4 for results.

The updated caption now reads:

*“**Figure 4.** Predictive fibers when including patients from both training cohorts (Cologne, Grenoble) irrespective of their target (top). Red fibers are positively associated with clinical improvement, blue fibers negatively. The sum of aggregated Fiber T-scores under each VTA explained %-Y-BOCS improvement in eight patients with 4 settings each (N = 32 stimulations) of the Madrid cohort (bottom left) and six patients of the London cohort with dual stimulation of STN and ALIC (bottom right). Please note that the p-values in this manuscript are based on random permutation testing. Based on classical tests, the result shown in the lower right panel would remain significant in a one-sided test, only (p -one-sided = 0.044, p -two-sided = 0.089). A replication of this result based on anatomically predefined pathways may be found in Figure S2.*

.” – p. 13, results.

372: “Data of such quality can usually not be acquired in individual patients and test-retest scan reliability in DBS settings has shown to be poor even when applying state-of-the-art MRI protocols”.

That statement is highly disputable. As far as I am aware, it has not been proven yet what the better concept might be: complete patient individual data sets (MRI, DWI with high quality fibertracking ability (multi-tensor, multi-shell), CT and VAT simulation) or normative approaches like using brain atlases, MRI templates, human connectome data from a different cohort etc.

In my point of view, this paper could non-arguably start such a discussion and promote as well as stimulate further studies and as such, I would like to welcome and invite the authors for further investigation in this field.

We agree and toned this statement down in the following:

“It may be logistically challenging to acquire data of such quality in a clinical routine setting (e.g. pre-operatively) in individual patients but will be feasible in specialized centers. Tractography based DBS targets pointed to coordinates that were sometimes >2 mm when repeating analyses on test-retest scans of the same subject (Petersen et al., 2017)⁸².” – discussion, p. 20-21

Naturally, we are very interested in the differences between normative and individualized connectomes, ourselves. In fact, we have just published a first comparative study as a preprint which shows high overlap between results but as expected, shows a trend of slightly better predictions when using individualized data (with no significant difference between measures). This study can be found here: <https://medrxiv.org/cgi/content/short/2020.02.24.20027490v1>.

By no way do we believe that normative connectomes are the final solution for connectomic surgery (and state so in the manuscript) – but we do believe we can still learn much by using them in the current state. Out-of-sample predictions and robustness across centers and cohorts may show that this is likely.

We have further adapted the discussion around this issue in the following paragraphs:

“Second, we used normative connectome data instead of patient-specific diffusion-weighted MRI data (which was unavailable for most of the patients included). This poses dramatic limitations given that such data cannot be representative of patient-specific anatomical variations. Still, we argue that some aspects about general pathophysiological mechanisms can still be investigated using normative data.” – discussion, p. 20

“However, patient-specific connectivity can never be reconstructed when using normative connectomes. Thus, normative connectomes will likely not embody the final solution to the connectomic surgery framework and will be challenged by advances in MRI technology and algorithm developments. Potentially, as a step in-between, using combined information from normative and patient-specific connectomes could embody a promising strategy that should be explored, in the future.” – discussion, p. 21

434: *For all patients in the three cohorts, high-resolution structural T1-weighted images were acquired on a 3.0-Tesla MRI-scanner, before surgery. Postoperative computer tomography (CT) was obtained in twenty-five patients after surgery to verify correct electrode placement, while eleven patients from the Grenoble cohort and the six London patients received postoperative MRI instead.*

This constitutes a problem: from a total of 42 patients only 25 received a post-op CT to confirm accurate location of the DBS electrodes and derive precise position of the individual DBS leads. The rest (17) received post-op MRI's. How were those data obtained? At 1.5T or at 3T? 3T is not clinically approved for DBS post-op DBS assessment while 1.5T MRI has relatively poor SNR to well define the target structures and provide high SNR data for good registration to the MNI templates.

Additionally how well could you define the DBS lead points based on MRI images that are

highly distorted in the vicinity of the electrodes. In my point of view this constitute a major flaw in the study that will lead to inconsistencies and congruent data processing outcome.

We thank the reviewer for raising this important issue. However, we will have to respectfully disagree on this issue. In our view, postoperative MRI has been a choice to evaluate electrode placements since a long time and two studies have even phantom-validated the approach based on 1.5T MRI (Yelnik et al., 2003, Journal of Neurosurgery; Pollo et al., 2004 Acta Neurochirurgica). The London group is known worldwide for relying on MRI for planning but also lead placement (Holl et al,2010, Neurosurgery; Foltynie et al, 2011, Neurosurgery & Psychiatry) and FDA-/CE-approved commercial systems (such as Medtronic SureTune 3 or Boston Guide XT) allow lead reconstructions based on postoperative MRI. In our own experience, reconstructions from CT or MRI lead to the same result in the same patients. We have larger samples of unpublished data but have published eight patients within the initial Lead-DBS methods paper (Horn & Kühn, 2015, NeuroImage). There, a mismatch of 0.66 mm was introduced by the choice of methods (in patients that received both postop CT and MRI) but this is early work from 2014 and we are confident that mismatches are even smaller using modern imaging protocols and analyses pipelines such as Lead-DBS v.2. Other studies have reported on similar agreements between modalities (Barnaure et al., 2015, Neuroradiology; Hyam et al., 2015, Neurosurgery; Xia et al., 2017, Journal of the Neurological Sciences).

We added the postoperative MRI information for the two samples in the methods section:

“Postoperative MRI parameters were as follows: Grenoble cohort: T1-weighted 3D-FFE scans were acquired on a 1.5T Philips MRI scanner with a 1.0x1.0x1.5 mm³ voxel size; TR: 20 ms, TE: 4.6 ms, flip angle: 30 deg. London cohort: T1-weighted 3D-MPRAGE scans were acquired on a 1.5T Siemens Espree interventional MRI scanner with a 1.5x1.5x1.5 mm³ voxel size and three-dimensional distortion corrected using the scanner’s built-in module; TR: 1410 ms, TE: 1.95 ms, FOV: 282 mm, flip angle: 10 deg, acquisition time 4 min and 32 s, relative SNR: 1.0.” – methods, p. 23-24

We still agree that this issue could constitute a limitation to our study and have added the following paragraph to the limitations section of our study:

“Besides, both post-operative CT (thirty-three patients) and post-operative MRI (seventeen patients) were used for electrode localization in the current dataset. Although studies have reported similar agreement between the results based on the two modalities, this might still lead to slight inconsistencies in the data. A larger dataset acquired with a homogeneous protocol would be ideal to validate our results in the future.” – discussion, p. 21

Changes to revised manuscript that are additional to reviewer comments:

During the review phase, we were able to reproduce main results of our study based on a basal ganglia tract atlas (Petersen et al. 2019 Neuron) that – in comparison to normative connectomes – is not prone to false positive connections. In contrast, the atlas is prone to false negative connections (tracts that are simply not realized in the atlas). By analyzing

results based on the tract atlas, we were able to pinpoint part of the connections we isolated from the normative connectome to represent the hyperdirect pathway connecting the dorsal anterior cingulate cortex to the STN. We do believe these additional analyses will help interpretation of our results and chose to include them in the revised manuscript. We discuss differences in interpretation of both the normative connectome and the tract atlas at length and show additional results as supplementary material. We wanted to briefly mention these additions to the manuscript here, as well.

Reviewers' Comments:

Reviewer #3:

Remarks to the Author:

Thank you for your revised version of the submitted manuscript and your comments to reviewer questions. I feel all questions and concerns have been adequately addressed and the paper has substantially gained in quality. I would therefore recommend the article for publication.